



# Marine isoprene production and consumption in the mixed layer of the surface ocean – A field study over 2 oceanic regions

Dennis Booge[1], Cathleen Schlundt[2], Astrid Bracher[3,4], Sonja Endres[1], Birthe Zäncker[1],

Christa A. Marandino[1]

[1]GEOMAR Helmholtz Centre for Ocean Research Kiel, Germany
[2]Marine Biological Laboratory, MBL, Woods Hole, MA, USA
[3]Alfred Wegener Institute - Helmholtz Centre for Polar and Marine Research, Bremerhaven, Germany
[4]Institute of Environmental Physics, University Bremen, Germany

*Correspondence to*: Dennis Booge (dbooge@geomar.de)

## Abstract

Parameterizations of surface ocean isoprene concentrations are numerous, despite the lack of source/sink process understanding. Here we present isoprene and related field measurements in the mixed layer from the Indian Ocean and the East Pacific Ocean to investigate the production and consumption rates in two contrasting regions, namely oligotrophic open ocean and coastal upwelling region. Our data show that the ability of different phytoplankton functional types (PFTs) to produce isoprene seems to be mainly influenced by light, ocean temperature, and salinity. Our field measurements also demonstrate that nutrient availability seems to have a direct influence on the isoprene production. With the help of pigment data, we calculate in-field isoprene production rates for different PFTs under varying biogeochemical and physical conditions. Using these new calculated production rates we demonstrate that an additional, significant and variable loss, besides a known chemical loss and a loss due to air sea gas exchange, is needed to explain the measured isoprene concentration. We hypothesize that this loss, with a lifetime for isoprene between 10 and 100 days depending on the ocean region, is attributed to heterotrophic respiration mainly due to bacteria.

## 1 Introduction

Isoprene (2-methyl-1,3-butadiene, $C_5H_8$), a biogenic volatile organic compound (VOC), accounts for half of the total global biogenic VOCs in the atmosphere (Guenther et al., 2012). 400-600 Tg C yr$^{-1}$ are emitted globally from terrestrial vegetation (Guenther et al., 2006;Arneth et al., 2008). Emitted isoprene influences the oxidative capacity of the atmosphere and acts as a source for secondary organic aerosols (SOA)(Carlton et al., 2009). It reacts with hydroxyl radicals (OH), as well as ozone and nitrate radicals (Atkinson and Arey, 2003;Lelieveld et al., 2008), forming low-volatility species, such as methacrolein or methyl vinyl ketone, which are then further photooxidized to SOA via more semi-volatile intermediate products (Carlton et al., 2009). Model studies suggest that isoprene accounts for 27% (Hoyle et al., 2007), 48% (Henze and Seinfeld, 2006) or up to 79% (Heald et al., 2008) of the total SOA production globally.

Whereas the terrestrial isoprene emissions are well known to act as a source for SOA, the oceanic source strength is highly discussed (Carlton et al., 2009). Marine derived isoprene emissions only account for a few percent of the total emissions and are suggested, based on model studies, to be generally lower than 1 Tg C yr$^{-1}$





(Palmer and Shaw, 2005;Arnold et al., 2009;Gantt et al., 2009;Booge et al., 2016). Some model studies suggest that these low emissions are not enough to control the formation of SOA over the ocean (Spracklen et al., 2008;Arnold et al., 2009;Gantt et al., 2009;Anttila et al., 2010;Myriokefalitakis et al., 2010). However, due to its short atmospheric lifetime of minutes to a few hours, terrestrial isoprene is not reaching the atmosphere over remote regions of the oceans. In these regions, oceanic emissions of isoprene could play an important role in SOA formation on regional and seasonal scales, especially in association with increased emissions during phytoplankton blooms (Hu et al., 2013). In addition, the isoprene SOA yield could be up to 29% under acid-catalyzed particle phase reactions during low-$NO_x$ conditions, which occur over the open oceans (Surratt et al., 2010). This SOA yield is significantly higher than a SOA burden of 2% during neutral aerosol experiments calculated by Henze and Seinfeld (2006).

Marine isoprene is produced by phytoplankton in the euphotic zone of the oceans, but only a few studies have directly measured the concentration of isoprene to date and the exact mechanism of isoprene production is not known. The concentrations range between $< 1$ and $200 \, \text{pmol L}^{-1}$ (Bonsang et al., 1992;Milne et al., 1995;Broadgate et al., 1997;Baker et al., 2000;Matsunaga et al., 2002;Broadgate et al., 2004;Kurihara et al., 2010;Zindler et al., 2014;Ooki et al., 2015;Hackenberg et al., 2017). Depending on region and season, concentrations of isoprene in surface waters can reach up to 395 and $541 \, \text{pmol L}^{-1}$ during phytoplankton blooms in the highly productive Southern Ocean and Arctic waters, respectively (Kameyama et al., 2014;Tran et al., 2013).

Studies have shown that the depth profile of isoprene mainly follows the chlorophyll-a (chl-a) profile suggesting phytoplankton as an important source (Bonsang et al., 1992;Milne et al., 1995;Tran et al., 2013) and furthermore, Broadgate et al. (1997) and Kurihara et al. (2010) could show a direct correlation between isoprene and chl-a concentrations in surface waters. However, this link is not consistent enough on global scales to predict marine isoprene concentrations using chl-a (Table 1). Laboratory studies with different monocultures illustrate that the isoprene production rate varies widely depending on the phytoplankton functional type (PFT) (Booge et al., 2016 and references therein). In addition, environmental parameters, such as temperature and light, have been shown to influence isoprene production (Shaw et al., 2003;Exton et al., 2013;Meskhidze et al., 2015). In general, the production rates increase with increasing light levels and higher temperature, similar to the terrestrial vegetation (Guenther et al., 1991). However, this trend cannot easily be generalized to all species, because each species-specific growth requirement is linked differently to the environmental conditions. For example, Srikanta Dani et al. (2017) showed that two diatom species, *Chaetoceros calcitrans* and *Phaeodyctylum tricornutum,* have their maximum isoprene production rate at light levels of 600 and $200 \, \mu\text{mol m}^{-2} \, \text{s}^{-1}$, respectively, which decreases at even higher light levels. Furthermore, Meskhidze et al. (2015) measured the isoprene production rates of different diatoms at different temperature and light levels on two consecutive days. Their results showed a less variable, but higher emission on day two, suggesting that phytoplankton must acclimate physiologically to the environment. This should also hold true for dynamic regions of the ocean and has to be taken into account when using field data to model isoprene production.

The main loss of isoprene in seawater is air-sea gas exchange, with a minor physical loss due to advective mixing and chemical loss by reaction with OH and singlet oxygen (Palmer and Shaw, 2005). The existence of biological losses still remains an open question, as almost no studies were conducted concerning this issue. Shaw et al. (2003) assumed the biological loss by bacterial degradation to be very small. However, Alvarez et al. (2009) showed that isoprene consumption in culture experiments from marine and coastal environments did not





exhibit first order dependency on isoprene concentration. They observed faster isoprene consumption with lower initial isoprene concentration.

This study significantly increases the small dataset of marine isoprene measurements in the world oceans with new observations of the distribution of isoprene in the surface mixed layer of the oligotrophic subtropical Indian Ocean and in the nutrient rich upwelling area of the East Pacific Ocean along the Peruvian coast. These two contrasting and, in terms of isoprene measurements, highly undersampled ocean basins are interesting regions to compare the diversity of isoprene producing species. With the help of concurrently measured physical

(temperature, salinity, radiation), chemical (nutrients, oxygen), and biological (pigments, bacteria) parameters, we aim to improve the understanding of isoprene production and consumption processes in the surface ocean under different environmental conditions.

## 2   Methods

### 2.1   Sampling sites

Measurements of oceanic isoprene were performed during three separate cruises, the SPACES (Science Partnerships for the Assessment of Complex Earth System Processes) and OASIS (Organic very short-lived substances and their air-sea exchange from the Indian Ocean to the stratosphere) cruises in the Indian Ocean and the ASTRA-OMZ (Air sea interaction of trace elements in oxygen minimum zones) cruise in the eastern Pacific Ocean. The SPACES/OASIS cruises took place in July/August 2014 on board the R/V Sonne I from Durban,

South Africa via Port Louis, Mauritius to Malé, Maldives and the ASTRA-OMZ cruise took place in October 2015 on board the R/V Sonne II from Guayaquil, Ecuador to Antofagasta, Chile (Figure 1).

### 2.2   Isoprene measurements

During all cruises, up to 7 samples (50 mL) from 5 to 150 m depth for each depth profile were taken bubble-free from a 24 L-Niskin bottle rosette equipped with a CTD (conductivity-temperature-depth; described in Stramma

et al. (2016)). Each vial contained 10 mL of helium headspace for purging. The water samples were, if necessary, stored in the fridge and analyzed on board, within 1 h of collection, using a purge and trap system attached to a gas chromatograph/mass spectrometer (GC/MS; Agilent 7890A/Agilent 5975C; inert XL MSD with triple axis detector) (Figure 2). Isoprene was purged for 15 minutes from the water sample with helium (70 mL min[-1]) containing 500 µL of gaseous deuterated isoprene (isoprene-d5) as an internal standard to account

for possible sensitivity drift (Figure 2: purge unit, load position). The gas stream was dried using potassium carbonate (SPACES/OASIS) or a Nafion® membrane dryer (Perma Pure; ASTRA-OMZ). $CO_2$- and hydrocarbon-free dry, pressurized air with a flow of 180 mL min[-1] was used as counter flow in the Nafion® membrane dryer (Figure 2: water removal). Before being injected into the GC (Figure 2: trap unit, inject position), isoprene was preconcentrated in a Sulfinert® stainless steel trap (1/16'' O.D.) cooled with liquid

nitrogen (Figure 2: trap unit, load position). The mass spectrometer was operated in single ion mode quantifying isoprene and d5-isoprene using m/z - ratios of 67, 68 and 72, 73, respectively. In order to perform daily calibrations for quantification, gravimetrically prepared liquid isoprene standards in ethylene glycol were diluted in Milli-Q water and measured in the same way as the samples. The precision for isoprene measurements was ± 8%.



### 2.3 Nutrient measurements

Micronutrient samples were taken on every cruise from the CTD bottles (covering all sampled depths). The samples from SPACES were stored in the fridge at -20°C and measured during OASIS. Samples from OASIS and ASTRA-OMZ were directly measured on-board with a QuAAtro auto-analyzer (Seal Analytical). Nitrate was measured as nitrite following reduction on a cadmium coil. The precision of nitrate measurements was calculated to be ±0.13 µmol $L^{-1}$.

### 2.4 Bacteria measurements

For bacterial cell counts, 4 mL samples were preserved with 200 µL glutaraldehyde (1% v/v final concentration) and stored at -20°C for up to three months until measurement. A stock solution of SybrGreen I (Invitrogen) was prepared by mixing 5 µL of the dye with 245 µL dimethyl sulfoxide (DMSO, Sigma Aldrich). 10 µL of the dye stock solution and 10 µL fluoresbrite YG microspheres beads (diameter 0.94 µm, Polysciences) were added to 400 µL of the thawed sample and incubated for 30 min in the dark. The samples were then analyzed at low flow rate using a flow cytometer (FACS Calibur, Becton Dickinson) (Gasol and Del Giorgio, 2000). TruCount beads (Becton Dickinson) were used for calibration and in combination with Fluoresbrite YG microsphere beads (0.5-1 µm, Polysciences) for absolute volume calculation. Calculations were done using the software program "Cell Quest Pro".

### 2.5 Phytoplankton groups from marker pigment measurements

Different PFTs were derived from marker phytoplankton pigment concentrations and chlorophyll concentrations. To determine PFT, 0.5 to 6 L of sea water were filtered through Whatman GF/F filters at the same stations isoprene was sampled. The soluble organic pigment concentrations were determined using high-pressure liquid chromatography (HPLC) according to the method of Barlow et al. (1997) adjusted to our temperature-controlled instruments as detailed in Taylor et al. (2011). We determined the list of pigments shown in Table 2 of Taylor et al. (2011) and applied the method by Aiken et al. (2009) for quality control of the pigment data. PFT was calculated using the diagnostic pigment analysis developed by Vidussi et al. (2001) and adapted in Uitz et al. (2006) to relate the weighted sum of seven, for each PFT representative diagnostic pigments (DP). By that the chl-a concentration for diatoms, dinoflagellates, haptophytes, chrysophytes, cryptophytes, cyanobacteria (excluding *Prochlorococcus* sp.), and chlorophytes were derived. The chl-a concentration of *Prochlorococcus* sp. was derived from the divinyl-chl-a concentration (marker pigment for this group) directly.

### 2.6 Photosynthetic available radiation within the water column measurements

Surface plane irradiance ($E_d(0^+,\lambda)$) data were taken from a RAMSES spectrometer and integrated from 400 to 700 nm to receive the downwelling photosynthetic available plane irradiance ($E_dPAR(0^+)$, in both units: W $m^{-2}$ and µmol $m^{-2}$ $s^{-1}$). The subsurface $E_dPAR(0^-)$ was calculated using the refractive index of water (n=1.34) and 0.98 for transmission assuming incident light angles <49°:

$$E_dPAR(0^-) = E_dPAR(0^+) \times 1.34^2 \times 0.98 \qquad (1)$$

In order to derive the diffuse attenuation coefficient ($K_d$) we calculated the euphotic depth ($Z_{eu}$) from the chl-a profile for all stations using the approximation by Morel and Berthon (1989) further refined by Morel and Maritorena (2001). In detail the following was done: From the chl-a profiles at each station the chl-a integrated





for $Z_{eu}$ ($C_{tot}$) was determined. A given profile was progressively integrated with respect to increasing depth ($z$). The successive integrated chl-a values were introduced in Equation 2 or 3 accordingly, thus providing successive $Z_{eu}$ values that were progressively decreasing. Once the last $Z_{eu}$ value, as obtained, became lower than the depth $z$ used when integrating the profile, these $C_{tot}$ and $Z_{eu}$ values from the last integration were taken. Profiles which

did not reach $Z_{eu}$ were excluded.

$$Z_{eu} = 912.5 \times C_{tot}^{-0.839} \; ; \text{if } 10m < Z_{eu} < 102m \tag{2}$$

$$Z_{eu} = 426.3 \times C_{tot}^{-0.547} \; ; \text{if } Z_{eu} > 102m \tag{3}$$

$K_d$ of each station was then calculated from $Z_{eu}$ as follows:

$$K_d = \frac{4.6}{Z_{eu}} \tag{4}$$

In order to derive the scalar photosynthetic available radiation at the surface ($PAR_{surface}$, µmol m$^{-2}$ s$^{-1}$) over the course of one day, $E_dPAR(0^-)$ one hourly averages were fitted with a sine function to account for the light variation during the day and converted into $PAR_{surface}$ by multiplying $E_dPAR(0^-)$ values with a factor of 2

(Jacovides et al., 2004) (Figure S1a shows an example for one day). The plane photosynthetic available irradiance at each depth ($z$) in the water column, $PAR(z)$, is then calculated applying Beer-Lambert's law (Figure S1b):

$$PAR(z) = PAR_{surface} \times e^{-K_d z}. \tag{5}$$

An example of two $E_dPAR(0+)$ fitted depth profiles is shown in the supplement (Figure S2).

### 2.7 Calculation of isoprene production

We calculated the isoprene production rate ($P$) in two different ways: a direct and an indirect calculation, which will be explained in the following paragraphs. For all calculations made we came up with one production rate per station within the mixed layer. This was either due to the shallow mixed layer depth (MLD) coming along with only one measurement within the mixed layer (coastal stations ASTRA-OMZ) or due to well mixed isoprene concentrations showing almost no gradient within the mixed layer (data explained in section 3.2).

#### 2.7.1    Direct calculation of isoprene production rates

Isoprene production rates of different PFTs were determined in laboratory phytoplankton culture experiments (see Table 2 in Booge et al. (2016)) and were used here to calculate isoprene production from measured PFTs in the field. These literature studies showed that isoprene production rates are light dependent, with increasing production rates at higher light levels (Shaw et al., 2003;Gantt et al., 2009;Bonsang et al., 2010;Meskhidze et al.,

2015). To include the light dependency in our calculations, we followed the approach of Gantt et al. (2009) for each PFT by applying a log squared fit between all single literature laboratory chl-a normalized isoprene production rates $P_{chloro}$ (µmol isoprene (g chl-a)$^{-1}$ h$^{-1}$) (references in Table 2) and their measured light intensity $I$ (µmol m$^{-2}$ s$^{-1}$) during individual experiments to determine an emission factor ($EF$) for each PFT (Figure S3):

$$P_{chloro} = EF \times \ln(I)^2 . \tag{6}$$

The resulting $EF$ from this log squared fit is unique for each PFT and is listed in Table 2: The higher the $EF$ of a

PFT, the higher its $P_{chloro}$ value at a specific light intensity. In order to calculate the isoprene production at each





station we used the scalar photosynthetic available radiation at each depth, *PAR(z)*, (see section 2.6) as input for
*I*, which was used with the respective, calculated *EF* of each PFT using Equation 6. The product was integrated
over the course of the day resulting in a $P_{chloro}$ value (µmol isoprene (g chl-a)$^{-1}$ day$^{-1}$) for each PFT and day
depending on the depth in the water column (Figure S4). The individual $P_{chloro,i}$ values of all PFTs were

multiplied with the corresponding, measured PFT concentration (*[PFT]$_i$*). The sum of all products gives the
directly calculated isoprene production rate ($P_{direct}$) for each station:

$$P_{direct} = \sum P_{chloro_i} \times [PFT]_i \,.\qquad(7)$$

### 2.7.2    Indirect calculation of isoprene production rates

The indirect calculation of the isoprene production rate is dependent on our measured isoprene concentrations
($C_{Wmeasured}$). We used the simple model concept of Palmer and Shaw (2005), assuming that the measured

isoprene concentration is in steady state, meaning that the production (*P*) is balanced by all loss processes:

$$P - C_{Wmeasured}\left(\sum k_{CHEM,i}C_{Xi} + k_{BIOL} + \frac{k_{AS}}{MLD}\right) - L_{MIX} = 0,\qquad(8)$$

where $k_{CHEM}$ is the chemical loss rate constant for all possible loss pathways (*i*) with the concentrations of the
reactants ($C_X$ = OH and $O_2$), $k_{BIOL}$ is the biological loss rate constant due to biological degradation, and $L_{MIX}$ is
the loss due to physical mixing. These constants are further described in Palmer and Shaw (2005). $k_{AS}$ is the loss
rate constant due to air-sea gas exchange scaled with the MLD. The MLD at each station was calculated from

CTD profile measurements applying the temperature threshold criterion (±0.2°C) of de Boyer Montégut et al.
(2004). $k_{AS}$ was computed using the Schmidt number ($S_C$) of isoprene (Palmer and Shaw, 2005) and the quadratic
wind-speed-based ($U_{10}$) parameterization of Wanninkhof (1992):

$$k_{AS} = 0.31\, U_{10}^2 \left(\frac{S_C}{660}\right)^{-0.5}.\qquad(9)$$

As we assume steady state isoprene concentration, we used the mean wind speed and the mean sea surface
temperature of the last 24 h before taking the isoprene sample to calculate $U_{10}$ and $S_C$, respectively.

We modified equation 8 to calculate the needed production rate ($P_{need}$) by multiplying $C_{Wmeasured}$ with the sum of
$k_{CHEM}$ (0.0527 day$^{-1}$) and $k_{AS}$ scaled with the MLD:

$$P_{need} = C_{Wmeasured}\left(k_{CHEM} + \frac{k_{AS}}{MLD}\right).\qquad(10)$$

We neglected the loss rates of isoprene due to biological degradation and physical mixing because they are low
compared to $k_{CHEM}$ and $k_{AS}$ (Palmer and Shaw, 2005;Booge et al., 2016), meaning that the resulting $P_{need}$ value
can be seen as a minimum needed production rate.

## 205  3    Results and discussion

### 3.1 Cruise settings

The first part of the Indian Ocean cruise, SPACES, started in Durban, travelled eastwards while passing the
Agulhas current and the southern tip of Madagascar (Toliara reef) with relatively warm water masses (mean:
23.4°C) and southerly winds. Southeast of Madagascar wind direction changed to easterly winds and we

encountered the Antarctic circumpolar current with significantly lower mean sea surface temperatures of 19.7°C
before heading north to Mauritius. Mean wind speed during the cruise was 8.2±3.7 m s$^{-1}$ and mean salinity was




35.5±0.2. Global radiation over the course of the day was on average ~360±70 W m$^{-2}$. As shown in Figure 3, within the mixed layer, chl-a concentrations were very low (average value < 0.3 µg L$^{-1}$) during the whole cruise, coinciding with generally low nutrient levels in the mixed layer (mean values for nitrate and phosphate were

0.14 and 0.15 µmol L$^{-1}$, respectively).

The second part of Indian ocean cruise, OASIS, covered open ocean regimes, upwelling regions, such as the equatorial overturning cell as described in Schott et al. (2009) and the shallow Mascarene Plateau (8°-12°S, 59°-62°E). Constant south easterly winds (mean: 10.3±4.2 m s$^{-1}$) were observed that were characteristic for the season of the southwest monsoon. During the cruise, sea surface temperature was constantly increasing with

latitude from 24.4°C (Port Louis) to 29.7°C (southern tip of the Maldives) with mean daily light levels of ~457±64 W m$^{-2}$. Salinity ranged from 34.4 to 35.4. As for the SPACES cruise, the chl-a concentration in the western tropical Indian Ocean was low (0.2-0.5 µg L$^{-1}$ on average, Figure 3). Nitrate levels (mean: 0.42 µmol L$^{-1}$) in the mixed layer were higher than during SPACES, but not phosphate (mean: 0.17 µmol L$^{-1}$).

The ASTRA-OMZ cruise took place in the coastal, wind driven Peruvian upwelling system (16°S - 6°S). This

area is a part of one of the four major eastern boundary upwelling systems (Chavez and Messié, 2009) and is highly influenced by the El Niño-Southern Oscillation. We observed constant southeasterly winds (8.2±2.5 m s$^{-1}$) travelling parallel to the Peruvian coast. During neutral surface conditions or La Niña conditions, cold, nutrient rich water is being upwelled at the shelf of Peru resulting in high biological productivity. However, in early 2015 a strong El Niño developed, which brought warmer, low salinity waters from the western Pacific to the coast of

Peru, resulting in suppressed upwelling with lower biological activity due to the presence of nutrient-poor water masses. The cruise started with a section passing the equator from north to south at 85.5°W east of the Galapagos Islands with mean sea surface temperatures of 25.0°C and low salinity waters (mean for profiles: 34.2), as well as low chl-a concentrations (mean for profiles: 0.5 µg L$^{-1}$). Levels of incoming shortwave radiation were ~508±67 W m$^{-2}$. Afterwards, we performed 4 onshore-offshore transects at about 9, 12, 14, and

16°S off the coast of Peru (Figure 1) where the incoming shortwave radiation was significantly decreased by clouds (~300 W m$^{-2}$). Upwelled waters identified by higher salinity (mean: 35.2) and lower sea surface temperatures (mean: 18.9°C) were found during the second part of the cruise. Chl-a values were highest directly at the coast (max: 13.1 µg L$^{-1}$), coinciding with lower sea surface temperatures (Figure 3) showing that some upwelling was still present.

### 3.2 Isoprene distribution in the mixed layer

The isoprene concentrations during the SPACES cruise were generally very low, ranging from 6.1 pmol L$^{-1}$ to 27.1 pmol L$^{-1}$ in the mixed layer (mean for the average of a profile: 12.3 pmol L$^{-1}$) in the southern Indian Ocean, mainly due to very low biological productivity. During the OASIS cruise, the isoprene concentrations south of 10°S were comparable to the concentrations of the SPACES cruise. North of 10°S, the isoprene values in the

mixed layer were significantly higher (mean: 35.9 pmol L$^{-1}$) (Figure 3). These results are in good agreement with the sea surface isoprene concentrations of Ooki et al. (2015) in the same area east of 60°E, who measured concentrations lower than 20 pmol L$^{-1}$ south of 12°S and concentrations of ~40 pmol L$^{-1}$ north of 12°S during a campaign between November 2009 and January 2010. During ASTRA-OMZ the concentrations ranged from 12.7 pmol L$^{-1}$ to 53.2 pmol L$^{-1}$ with a mean isoprene concentration of 29.5 pmol L$^{-1}$ in the mixed layer. Although

the chl-a concentrations at the coastal stations (3.8 µg L$^{-1}$) were significantly higher than open ocean values (0.7 µg L$^{-1}$), the isoprene values did not show the same trend (Figure 3).





A mean normalized depth profile of each cruise for isoprene (blue), water temperature (black), oxygen (red), and chl-a (green) is shown in Figure 4. In order to compare the depth profiles of each cruise with respect to the different concentration regimes, we normalized the measured values by dividing the mean concentration in the

mixed layer of each station by the concentration of each depth from the same station profile. A normalized value >1 means that the value at a certain depth is higher than the mean value in the mixed layer, a value <1 means less than in the mixed layer. As the sampled depths at each station were not the same at every cruise, we binned the data into seven equally spaced depth intervals (15 m) and averaged each data of an interval over each of the three cruises. The calculated mean mixed layer depths of the SPACES and OASIS cruises, using the temperature

threshold criterion (±0.2°C) of de Boyer Montégut et al. (2004), were about 60 m, the mean mixed layer depth of the ASTRA-OMZ cruise was 30 m excluding the four coastal stations, which had only a MLD of 20 m resulting in only one bin interval in the MLD. Figure 4 shows, that during all three cruises almost no gradient of isoprene in the mixed layer was detectable. In contrast to the isoprene concentration, the highest chl-a concentration was measured slightly above or below the MLD during SPACES/OASIS, whereas during ASTRA-OMZ chl-a

showed the same trend as isoprene. These results suggest a very fast mixing of isoprene after it is produced by phytoplankton and released to the water column above the MLD.

As isoprene is produced biologically by phytoplankton, many studies attempted to find a correlation between chl-a and isoprene, but found very different results. Bonsang et al. (1992), Milne et al. (1995) and Zindler et al. (2014) did not find a significant correlation, whereas other studies could show a significant correlation and,

therefore, attempted a linear regression to show a relationship between isoprene and chl-a, as well as SST (Broadgate et al., 1997;Kurihara et al., 2010;Kurihara et al., 2012;Ooki et al., 2015;Hackenberg et al., 2017). Comparing the different factors of each regression equation (Table 1), it can be seen that there is no globally unique relationship between chl-a (and SST) and isoprene. As shown in Table 1, during ASTRA-OMZ there was no significant correlation between chl-a and isoprene, whereas during SPACES and OASIS the correlation was

significant but with low $R^2$-values (SPACES: $R^2$=0.30, OASIS: $R^2$=0.10) and different regression coefficients. Hackenberg et al. (2017) split their data from three different cruises into two SST bins with SST values higher and lower than 20°C, resulting in significant correlations with $R^2$-values from 0.37 to 0.82 depending on the cruise (Table 1). Ooki et al. (2015) described a multiple linear relationship between isoprene, chl-a and SST when using three different SST regimes (Table 1). Our correlations, using the approaches of Ooki et al. (2015)

and Hackenberg et al. (2017), were significant, except for SST values higher than 27°C, but the regression coefficients were also significantly different to those found by Ooki et al. (2015) and Hackenberg et al. (2017). These varying equations demonstrate that bulk chl-a concentrations, or linear combinations of chl-a concentration and SST, do not adequately predict the variability of isoprene in the global surface ocean, but do point to these variables as among the main controls on isoprene concentration in the euphotic zone.

**3.3 Modeling chl-a normalized isoprene production rates**

The directly calculated production rate ($P_{direct}$) using Equation 7 and the indirectly calculated production rate ($P_{need}$) using Equation 10 were compared and were found to be significantly different (Figure 5a, difference in percent: ($P_{direct}$ - $P_{need}$)/$P_{need}$*100). The difference of more than -60% between $P_{direct}$ and $P_{need}$ during SPACES/OASIS means that $P_{direct}$ is too low to account for the measured isoprene concentrations, which is also

true for the equatorial region of ASTRA-OMZ. In the open ocean region of ASTRA-OMZ, the average difference between $P_{direct}$ and $P_{need}$ is the lowest but still highly variable from station to station. However, in the



coastal region of ASTRA-OMZ the directly calculated isoprene production rate is highly overestimating the needed production by 75% on average. There are two possible explanations for this difference: 1) the presence of a missing sink, which is not accounted for in the calculation of $P_{need}$. Adding an additional loss term to equation

10 would increase the needed production to reach the measured isoprene concentration. This sink would only be valid for this specific coastal region, but would increase the discrepancy between $P_{direct}$ and $P_{need}$ for all other performed cruises. Furthermore, this possible loss rate constant would have to be on average 0.22 day$^{-1}$ and, therefore, higher than the main loss due to air sea gas exchange in the coastal region (see section 3.5 and Figure 8). Thus, it is highly unlikely that this additional loss term is the only reason for the discrepancy between $P_{direct}$

and $P_{need}$; 2) incorrect literature derived chl-a normalized isoprene production rate ($P_{chloro}$) for one or more groups of PFTs. For example, the high $P_{direct}$ values, compared to the $P_{need}$ values, during ASTRA-OMZ coincided with high chl-a concentrations in the coastal area. These coastal stations were, in contrast to all other measured stations, highly dominated by diatoms (up to 7.67 µg L$^{-1}$, Figure S5). This might point to a possibly incorrect $P_{chloro}$ value (too high) for diatoms (and other PFTs).

Therefore, we calculated new individual chl-a normalized production rates of each PFT ($P_{chloronew}$). We used the concentrations of haptophytes, cyanobacteria and *Prochlorococcus* for SPACES/OASIS and the concentrations of haptophytes, chlorophytes and diatoms for ASTRA-OMZ, as these PFT were the three most abundant PFTs of each cruise, accounting on average for ≥80% of total PFTs. We performed a multiple linear regression by fitting a linear equation between the $P_{need}$ values for each station and the corresponding PFT concentrations (analogous

to equation 7) to derive one new calculated $P_{chloronew}$ value for each PFT and cruise, which is listed in Table 3. The lower and upper limit of the $P_{chloronew}$ value was set to 0.5 and 50 µmol (g chl-$a$)$^{-1}$ day$^{-1}$, respectively, when performing the multiple linear regression, to avoid mathematically possible but biologically unreasonable negative chl-a normalized isoprene production rates. The upper limit was chosen in relation to the maximum published chl-a normalized isoprene production rate of *Prasinococcus capsulatus* by Exton et al. (2013)

(32.16±5.76 µmol (g chl-a)$^{-1}$ day$^{-1}$). This rate was measured during common light levels of 300 µmol m$^{-2}$ s$^{-1}$. Applying a same log squared relationship between light levels and the isoprene production rate as for the other PFTs would increase this value up to 50 µmol (g chl-$a$)$^{-1}$ day$^{-1}$ at light levels of ~1000 µmol m$^{-2}$ s$^{-1}$. We only used the three most abundant PFTs for each cruise, which, contribute on average ≥80% to the total phytoplankton chl-a concentration. Our tests using the whole PFT community for the multiple linear regression

did not change our results and, in some cases, led to highly unlikely production rates for the less abundant PFTs. With the help of the multiple linear regression derived $P_{chloronew}$ values, we calculated the new direct isoprene production rate ($P_{calc}$) in the same way as $P_{direct}$ in equation 7. We compared our calculated $P_{calc}$ values with the $P_{need}$ values, which are shown in Figure 5b (difference in percent between $P_{calc}$ and $P_{need}$). We found one outlier station for each cruise (SPACES: Station 1, OASIS: Station 10, ASTRA-OMZ: Station 17), when using the new

$P_{chloronew}$ values for each PFT for each whole cruise (Figure 5b, left part). We excluded these stations from every following calculation and redid the multiple linear regression. Furthermore, we split the ASTRA-OMZ into three different regions (equator, coast and open ocean), due to their contrasting biomass to isoprene concentration ratio, and calculated new $P_{chloronew}$ values for each of the three most abundant PFTs for SPACES, OASIS, and each part of ASTRA-OMZ.

Haptophytes were one of the three most abundant PFTs during all three cruises (Figure S5) and their $P_{chloronew}$ values range from 0.5 to 47.9 µmol (g chl-$a$)$^{-1}$ day$^{-1}$ with a mean value of 17.9 ± 18.3 µmol (g chl-$a$)$^{-1}$ day$^{-1}$ for all cruises. The haptophyte production rates exhibited two interesting features. First, this range is highly variable





depending on the oceanic region (tropical ocean (SPACES), subtropical ocean (OASIS)) and different ocean regimes (coastal, open ocean). Second, the average value is different than the mean value of all laboratory study

derived isoprene production rates of haptophytes ($6.92\pm5.78$ µmol (g chl-$a$)$^{-1}$ day$^{-1}$, Table 3). During SPACES/OASIS the $P_{chloronew}$ values of *Prochlorococcus* (both 0.5 µmol (g chl-$a$)$^{-1}$ day$^{-1}$) are lower than the mean literature value (9.66 µmol (g chl-$a$)$^{-1}$ day$^{-1}$, Table 3), whereas the cyanobacteria values are higher (44.7 and 13.9 µmol (g chl-$a$)$^{-1}$ day$^{-1}$) than the literature value (6.04 µmol (g chl-$a$)$^{-1}$ day$^{-1}$, Table 3). Chlorophytes, as well as diatoms, are known to be low isoprene producers with mean $P_{chloro}$ values of 1.47 µmol (g chl-$a$)$^{-1}$ day$^{-1}$

and 2.54 µmol (g chl-$a$)$^{-1}$ day$^{-1}$, respectively (Table 3). For diatoms, this is verified with our calculated rates during ASTRA-OMZ (all values $\leq 0.6$ µmol (g chl-$a$)$^{-1}$ day$^{-1}$), whereas the rate for chlorophytes in the coastal regions (6.1 µmol (g chl-$a$)$^{-1}$ day$^{-1}$) is significantly higher than in the open ocean and equatorial region during ASTRA-OMZ (0.5 µmol (g chl-$a$)$^{-1}$ day$^{-1}$). Over all three cruises no significant correlations were found between the new multiple linear regression derived $P_{chloronew}$ values of each PFT and any other parameter measured on the

cruise. This may be caused by the high variability of the chl-a normalized production rates of different PFTs (Table 3). Another explanation could be the high variability of isoprene production of different species within one PFT group. For instance, in the PFT group of haptophytes, the isoprene production rates of two different strains of *Emiliania huxleyi* measured by Exton et al. (2013) were $11.28 \pm 0.96$ and $2.88 \pm 0.48$ µmol (g chl-$a$)$^{-1}$ day$^{-1}$ for strain CCMP 1516 and CCMP 373, respectively. Laboratory culture experiments show that stress

factors, like temperature and light, also influence the emission rate within one species (Shaw et al., 2003;Exton et al., 2013;Meskhidze et al., 2015). Srikanta Dani et al. (2017) showed that in a light regime of 100-600 µmol m$^{-2}$ s$^{-1}$ the isoprene emission rate was constantly increasing with higher light levels for the diatom *Chaetoceros calcitrans*, whereas the diatom *Phaeodyctylum tricornutum* was highest at 200 µmol m$^{-2}$ s$^{-1}$ and decreased at higher light levels. Furthermore, health conditions (Shaw et al., 2003), as well as the growth stage

of the phytoplankton species (Milne et al., 1995), can also influence the isoprene emission rate.

With the new $P_{calc}$ values, we slightly overestimate the needed production $P_{need}$ by up to 20% on average (Figure 5b, right part). For SPACES and OASIS, except for station 1 and 10, using one $P_{chloronew}$ value for each PFT for the whole cruise is reasonable because the biogeochemistry in these regions did not differ much within one cruise. This was not true for ASTRA-OMZ, due to the biogeochemically contrasting open ocean region and the

coastal upwelling region. Using just one $P_{chloronew}$ value for each PFT for the whole cruise resulted in a highly overestimated and variable $P_{calc}$ value (Figure 5b, "ASTRA-OMZ"). Therefore splitting this cruise into three different parts (equator, coast, open ocean), due to their different chl-$a$ concentration and nutrient availability, resulted in less variable $P_{calc}$ values. However, in the coastal region the variability is still the highest, but with the new derived $P_{calc}$ the agreement with $P_{need}$ is significantly better than with $P_{direct}$ (compare Figure 5a and b).


### 3.4 Drivers of isoprene production

As mentioned above, no significant correlations between each calculated $P_{chloronew}$ value and any other parameter during the three cruises were found. However, comparing the calculated isoprene production rates of the haptophytes with global radiation, ocean temperature, salinity and nitrate results in some interesting qualitative

trends (Figure 6). Mean global radiation during SPACES (~360 W m$^{-2}$) was lower than during OASIS (~457 W m$^{-2}$). Highest mean values were measured during ASTRA-OMZ (~508 W m$^{-2}$). The same trend can be seen in the $P_{chloronew}$ values of the haptophytes. Within the open ocean and coastal regimes of ASTRA-OMZ, the





isoprene production rate was low, again showing the same trend as the mean global radiation (decreased to
~310 W m$^{-2}$). A similar trend can be seen with the mean ocean temperature and the $P_{chloronew}$ values of the

haptophytes. These results are similar to several laboratory experiments with monocultures: Higher light
intensities and water temperatures enhance phytoplankton ability to produce isoprene (Shaw et al., 2003;Exton et
al., 2013;Meskhidze et al., 2015). However, Meskhidze et al. (2015) showed in laboratory experiments that
isoprene production rates from two diatoms species were highest when incubated in water temperatures of 22 to
26°C. Higher temperatures caused a decrease in isoprene production rate. During OASIS, mean water

temperatures were 27.3°C with up to 29.2°C near the Maldives. If this temperature dependence can be
transferred from diatoms also to haptophytes, the high seawater temperatures during OASIS could explain why
the calculated isoprene production rate is lower than in the ASTRA-OMZ-equatorial regime. Another reason for
the very high isoprene production rate of haptophytes in the equatorial regime during ASTRA-OMZ, apart from
temperature and light intensity, could be stress-induced production caused by low saline waters, which was

already shown for dimethylsulphoniopropionate, a precursor for the climate relevant trace gas dimethyl sulphide,
produced by phytoplankton (Shenoy et al., 2000). The salinity is considerably lower at the equator during
ASTRA-OMZ than for all other cruise regions, with values down to 33.4. We observed that the $P_{chloronew}$ values
decrease again in regions with higher saline waters, where phytoplankton likely experience less stress due to
salinity, temperature or light levels.

In order to identify parameters that influence not only the chl-a normalized isoprene production rate of
haptophytes, but the rate of all PFTs together, we calculated a normalized isoprene production rate ($P_{norm}$)
independent from the absolute amount of each PFT. Hence, we divided each $P_{calc}$ value at every station by the
amount of the three most abundant PFTs:

$$P_{norm} = \frac{\sum_{i=1}^{3} P_{chloronew_i} \times [PFT]_i}{\sum_{i=1}^{3} [PFT]_i} = \frac{P_{calc}}{\sum_{i=1}^{3} [PFT]_i} \qquad (11)$$

i = three most abundant PFTs during each cruise.

The $P_{norm}$ value helps us to obtain more insight about the influencing factors at each station, rather than only one
mean data point for each cruise. We plotted the $P_{norm}$ values of each station versus the ocean temperature and
color coded them by nitrate concentration as a marker for the nutrient availability (Figure 7). During SPACES
(squares) and OASIS (triangles), the normalized production rate is on average 12.8±2.2 pmol (μg PFT)$^{-1}$ day$^{-1}$
and independent from the ocean temperature, while the nitrate concentration is very low (0.33±0.53 μmol L$^{-1}$).

During ASTRA-OMZ (circles) in the coastal and open ocean region, the nitrate concentrations were significantly
higher (16.4±5.5 μmol L$^{-1}$), but the $P_{norm}$ values were lower (< 8 pmol (μg PFT)$^{-1}$ day$^{-1}$) correlating with lower
ocean temperatures. In the equatorial region of ASTRA-OMZ, the production rates are significantly higher than
during SPACES and OASIS, with up to 36.4 pmol (μg PFT)$^{-1}$ day$^{-1}$. On the right panel of Figure 7, the mean
salinity for each $P_{norm}$ dependent box (separated by the dashed lines) is shown. ASTRA-OMZ (equator) and

SPACES and OASIS do not differ in ocean temperature or in nitrate concentration. However, the normalized
production is significantly higher at the ASTRA-OMZ equatorial region, which may be caused by the low
salinity there. In summary: 1) During ASTRA-OMZ (coast, open ocean) $P_{norm}$ is comparably lower
(< 8 pmol (μg PFT)$^{-1}$ day$^{-1}$) under "biogeochemically active" conditions (high nitrate concentration) but
increases with increasing ocean temperature, 2) Under limited nutrient conditions $P_{norm}$ is significantly increased

likely due to nutrient stress 3) If the phytoplankton are additionally stressed due to lower salinity, $P_{norm}$ is
furthermore increased. These results show that there is no main parameter driving the isoprene production rate,





resulting in a more complex interaction of physical and biological parameters influencing the phytoplankton to produce isoprene.

### 3.5 Loss processes

The comparison between $P_{calc}$ and $P_{need}$ in Figure 5b shows a mean overestimation of 10-20%. This is likely due to a missing loss term in the calculation, which would balance out the needed and calculated isoprene production. Chemical loss (red dashed line) and loss due to air sea gas exchange (black solid line) using the gas transfer parameterization of Wanninkhof (1992) were already included in the calculation (Equation 10) and their loss rate constants are shown in Figure 8. For comparison, we added the $k_{AS}$ values using the parameterizations

of Wanninkhof and McGillis (1999) (black dotted line) and Nightingale et al. (2000) (black dashed line). They have different wind speed dependencies of gas transfer, which could influence the computed isoprene loss at high wind speeds. The parameterization of Wanninkhof and McGillis (1999) is cubic and will increase the loss rate constant of isoprene due to air sea gas exchange at high winds compared to the other parameterizations (Figure 8, OASIS). Nightingale et al. (2000) is a combined linear and quadratic parameterization, which would

decrease the isoprene loss due to air sea gas exchange. However, during these cruises the wind speed was between 8 and 10 m s$^{-1}$ where the parameterization of Wanninkhof (1992) is higher than both Wanninkhof and McGillis (1999) and Nightingale et al. (2000). Therefore the use of these alternative parameterizations would even lower the loss rate constant due to air sea gas exchange, leading to the need of an additional loss rate in order to balance the isoprene production.

To calculate the additionally required consumption rate ($k_{consumption}$), we only used stations where a loss term was actually needed to balance the calculated and needed production ($P_{calc} > P_{need}$). Those values were averaged within each cruise and are shown in Figure 8. For comparison, we added the loss rate constants due to bacterial consumption from Palmer and Shaw (2005) (blue dashed line; 0.06 day$^{-1}$) and an updated value from Booge et al. (2016) (blue dotted line; 0.01 day$^{-1}$). Comparable to the chemical loss rate, the $k_{BIO}$ values were assumed to be

constant (following the assumption of Palmer and Shaw (2005)), because no data about bacterial isoprene consumption in surface waters is available. Figure 8 clearly shows that the needed loss rate constant is not a constant factor. During SPACES and OASIS the loss rate constant is roughly in the middle of the assumed $k_{BIO}$ values of Palmer and Shaw (2005) and Booge et al. (2016), whereas during ASTRA-OMZ (equator and open ocean) the calculated loss rate constant fits quite well with the assumed value of Booge et al. (2016). In all four

regions, the additional calculated sink is lower than the chemical loss and the loss due to air sea gas exchange, which is not true for the coastal region of ASTRA-OMZ. The loss rate constant (0.1 day$^{-1}$) is about 10 times higher than in the open ocean region, resulting in a lifetime of isoprene of only 10 days, which is comparable to the lifetime due to air sea gas exchange during SPACES and OASIS. Physical loss, like advective mixing through the thermocline, cannot account for this sink, as this lifetime is assumed to be several years (Palmer and

Shaw, 2005) and, therefore, negligible. Even a change in the chemical loss rate would only change the absolute value of the calculated loss rate constant, but not its variability. We tested a temperature dependent rate for the reaction with OH, but the mean difference of the temperature dependent $k_{CHEM}$ to the non-temperature dependent $k_{CHEM}$ was less than 2% for all temperature regimes during the cruises and, therefore, negligible. It must be noted that the loss rates due to reactions with OH and singlet oxygen are gas phase reaction rates, meaning that they

might not be suitable for reactions in the water phase. These rates, involving possible temperature and pressure dependencies, have to be evaluated in water in order to determine the chemical loss in the water column.





Marine produced halocarbons, like dibromomethane and methyl bromide, are known to undergo bacterial degradation (Goodwin et al., 1998). Compared to halocarbons isoprene is not toxic and has two energy-rich double bonds and, therefore, may be even favored to be oxidized by heterotrophic marine bacteria (Alvarez et

al., 2009). Figure 9 shows a comparison of total bacteria counts and isoprene concentration from each station in the MLD. The correlation between bacteria and the concentration of isoprene is only significant when haptophytes are less than 33% of the total phytoplankton chl-a concentration ($R^2$=0.80, p=2.34*$10^{-7}$). Haptophytes were one of the three dominant PFTs during all cruises and had a mean calculated isoprene production rate of 17.9 µmol (g chl-$a$)$^{-1}$ day$^{-1}$ (Table 3). Compared to literature values of other PFTs this is a

high isoprene production rate. Multiplying this value with the chl-a concentration of haptophytes results in a mean isoprene production rate of ~ 3 pmol L$^{-1}$ day$^{-1}$ which is about 4 times higher than the mean calculated loss rate due to bacterial degradation over all cruises (~ 0.8 pmol L$^{-1}$ day$^{-1}$). This leads to the hypothesis that, if the phytoplankton community is dominated (>33%) by haptophytes, the isoprene production rate is much higher than the degradation rate by bacteria and, therefore, no longer correlated to the bacteria abundance.

Due to the different loss rate constants of bacterial degradation (~0.01 day$^{-1}$ during ASTRA-OMZ (equator) compared to ~0.1 day$^{-1}$ in the coastal region of ASTRA-OMZ, Figure 8) in the different regions it is important to scale the loss. Unfortunately, the absolute amount of bacteria does not have a significant influence on $k_{consumption}$ (Figure 10a,b), which may be caused by different heterotrophic bacteria with a different ability to use isoprene as an energy source. However, we find a similar qualitative trend for $k_{consumption}$ and the apparent oxygen utilization

(AOU) (difference of equilibrium oxygen saturation concentration and the actual measured dissolved oxygen concentration) during the three cruises (Figure 10c). The higher loss rate constant of isoprene due to possible bacterial consumption coincides with considerably higher AOU values in the coastal regime of ASTRA-OMZ, which may be caused by heterotrophic respiration. Even if this correlation is not significant, this trend points to the influence of environmental conditions on biological activity, which in turn influences the isoprene

consumption.

## 4    Conclusions

For the first time, marine isoprene measurements were performed in the eastern Pacific Ocean. In addition, our isoprene measurements in the highly undersampled Indian Ocean further increase the small dataset of oceanic isoprene measurements in this region. The results from both oceans show that isoprene is well mixed in the

MLD. Despite the known biogenic origin of isoprene, the marine isoprene concentrations cannot be described globally with a simple parameterization including chl-a concentration or SST or a combination of both. On regional scales this relationship might be sometimes significant (Ooki et al., 2015;Hackenberg et al., 2017), but laboratory monoculture experiments show that isoprene production rates range widely over all different PFTs, as well as within one PFT (Booge et al., 2016 and references therein). The production rates from laboratory

experiments have to be evaluated in the field, as different PFTs are not distributed equally over the world ocean and are also influenced by temperature and salinity, as well as changing light levels. Therefore we used isoprene measurements as well as different phytoplankton marker pigment measurements to derive in-field productions rates for haptophytes, cyanobacteria, *Prochlorococcus*, chlorophytes, and diatoms in different regions. The results show that the isoprene production is influenced by light, ocean temperature, and salinity, with an

indication that the nutrient regime might exert some influence. Our calculations also show that, besides chemical



loss and the loss due to air sea gas exchange, another non-static isoprene consumption process has to be taken into account to understand isoprene concentrations in the surface ocean. This loss may be attributed to bacterial degradation, or more generally, to heterotrophic respiration, as we could show a similar qualitative trend between the additional loss rate constant and the AOU. These results clearly indicate that further experiments are

needed to evaluate isoprene production rates for every PFT in general, but additionally under different biogeochemical conditions (light, salinity, temperature, nutrients). With the help of incubation experiments under different conditions, the additional loss process can be investigated. The exact knowledge of the different production and loss processes, as well as their interaction, is crucial in understanding global marine isoprene cycling. Air sea gas exchange, the main loss process for isoprene in the ocean, has further to be assessed due to

the variability and the uncertainty of the different k-parameterizations. Different parameterizations under different wind levels highly influence the loss term, which is additionally influenced by surface films at low or bubble generation at high wind speeds. The evaluation of these loss processes, in conjunction with the complex variability of production by phytoplankton, should be further examined in order to predict marine isoprene concentrations and evaluate its impact on SOA formation over the remote open ocean.

## 5    Data availability

All isoprene data and bacterial cell counts are available from the corresponding author. Pigment and nutrient data from SPACES/OASIS and ASTRA-OMZ will be available from PANGAEA, but for now can be obtained through the corresponding author.

**Acknowledgements**

The authors would like to thank the captain and crew of the R/V Sonne during SPACES/OASIS and ASTRA-OMZ, as well as the chief scientist Kirstin Krüger (SPACES/OASIS). We thank Sonja Wiegmann for HPLC pigment analysis of SPACES/OASIS and ASTRA-OMZ samples, Sonja Wiegmann and Wee Cheah for pigment sampling during SPACES/OASIS, Rüdiger Röttgers for helping with pigment sampling and radiation measurement during ASTRA-OMZ, Tania Klüver for flow cytometry analysis, and Martina Lohmann for

nutrient sampling and analysis during SPACES/OASIS and ASTRA-OMZ. The authors gratefully acknowledge NASA for providing the satellite MODIS-Aqua data. This work was carried out under the Helmholtz Young Investigator Group of Christa A. Marandino, TRASE-EC (VH-NG-819), from the Helmholtz Association through the President's Initiative and Networking Fund and the GEOMAR Helmholtz-Zentrum für Ozeanforschung Kiel. The R/V Sonne I cruises SPACES/OASIS and R/V Sonne II cruise ASTRA-OMZ were

financed by the BMBF through grants 03G0235A and 03G0243A, respectively.

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

**Table 1: Factors of different regression equations ([isoprene]=u*[chl-a]+v*SST+intercept) from different studies compared to factors from this study. Bold/*italic* $R^2$ value: correlation significant/not significant (significant: $p<0.05$).**
**[chl-a] in µg L$^{-1}$, SST in °C, [isoprene] in pmol L$^{-1}$.**

| reference | cruise/region | SST bins | u | v | intercept | R² |
|---|---|---|---|---|---|---|
| Hackenberg et al. (2017) | AMT 22 (Atlantic O.) | <20°C | 37.9 | --- | 17.5 | **0.37** (n=39) |
| | AMT 23 (Atlantic O.) | | 15.1 | --- | 18.4 | **0.55** (n=11) |
| | ACCACIA 2 (Arctic) | | 34.1 | --- | 11.1 | **0.61** (n=34) |
| | AMT 22 (Atlantic O.) | ≥20°C | 300 | --- | -3.35 | **0.60** (n=93) |
| | AMT 23 (Atlantic O.) | | 103 | --- | 5.58 | **0.82** (n=22) |
| Ooki et al. (2015) | Southern Ocean, Indian Ocean, Northwest Pacific Ocean, Bering Sea, western Arctic Ocean | 3.3-17°C | 14.3 | 2.27 | 2.83 | 0.64 |
| | | 17-27°C | 20.9 | -1.92 | 63.1 | 0.77 |
| | | >27°C | 319 | 8.55 | -244 | 0.75 |
| Kurihara et al. (2012) | Sagami Bay | no bin | 10.7 | --- | 5.9 | 0.49 (n=8) |
| Kurihara et al. (2010) | Western North Pacific | no bin | 18.8 | --- | 6.1 | 0.79 (n=60) |
| Broadgate et al. (1997) | North Sea | no bin | 6.4 | --- | 1.2 | 0.62 |
| This study | whole study | no bin | 2.45 | --- | 22.1 | **0.07** (n=138) |
| | SPACES (Indian Ocean) | | 20.2 | --- | 8.01 | **0.30** (n=37) |
| | OASIS (Indian Ocean) | | 42.6 | --- | 12.6 | **0.10** (n=59) |
| | ASTRA-OMZ (Southeast Pacific O.) | | 1.26 | --- | 26.5 | *0.07* (n=42) |
| | | <20°C | 3.92 | --- | 11.5 | **0.59** (n=46) |
| | | ≥20°C | 25.6 | --- | 16.6 | **0.14** (n=92) |
| | | 3.3-17°C | 1.30 | 10.0 | -144 | **0.84** (n=10) |
| | | 17-27°C | 10.4 | 0.76 | -3.70 | **0.41** (n=97) |
| | | >27°C | 40.4 | -0.58 | 39.7 | *0.17* (n=31) |





**Table 2: Emission factor (*EF*) of each PFT determined by applying a log squared relationship between light intensity and isoprene production rates resulting from published phytoplankton cultures experiments.**

| Species | emission factor | references of literature values used for fitting |
|---|---|---|
| Diatoms | 0.0064 | Shaw et al. (2003), Bonsang et al. (2010), Exton et al. (2013), Meskhidze et al. (2015) |
| Chlorophytes | 0.0168 | Shaw et al. (2003), Bonsang et al. (2010), Exton et al. (2013) |
| Dinoflagellates | 0.0176 | Exton et al. (2013) |
| Haptophytes | 0.0099 | Shaw et al. (2003), Bonsang et al. (2010), Exton et al. (2013) |
| Cyanobacteria | 0.0097 | Shaw et al. (2003), Bonsang et al. (2010), Exton et al. (2013) |
| Cryptophytes | 0.0120 | Exton et al. (2013) |
| *Prochlorococcus* | 0.0053 | Shaw et al. (2003) |


**Table 3: Calculated chl-a normalized isoprene production rates ($P_{chloronew}$, µmol (g chl-*a*)$^{-1}$ day$^{-1}$) of the three most abundant PFTs during SPACES/OASIS (haptophytes, cyanobacteria, *Prochlorococcus*) and ASTRA-OMZ (haptophytes, chlorophytes, diatoms). Number indicated after \ denotes that a station that has been excluded from the analysis. For explanation of the omission, please refer to paragraph 3.3.**


| cruise | | haptophytes | cyanobacteria | *Prochlorococcus* | chlorophytes | diatoms |
|---|---|---|---|---|---|---|
| SPACES\1 | | 0.5 | 44.7 | 0.5 | -- | -- |
| OASIS\10 | | 21.2 | 13.9 | 0.5 | -- | -- |
| **ASTRA -OMZ** | equator | 47.9 | -- | -- | 0.5 | 0.5 |
| | coast\17 | 9.6 | -- | -- | 6.1 | 0.6 |
| | open ocean | 10.3 | -- | -- | 0.5 | 0.5 |
| Literature values Booge et al. (2016) | | 6.92 | 6.04 | 9.66 | 1.47 | 2.54 |



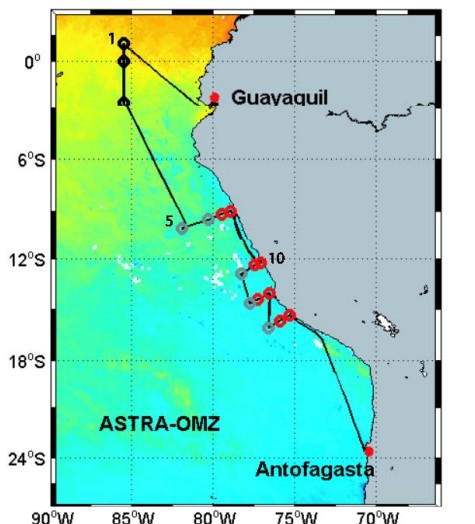
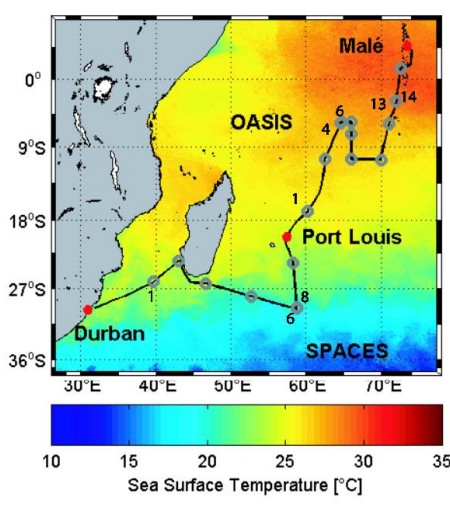

**Figure 1: Cruise tracks (black) of ASTRA-OMZ (October 2015, East Pacific Ocean) and SPACES/OASIS (July/August 2014, Indian Ocean) plotted on top of monthly mean sea surface temperature detected by the Moderate Resolution Imaging Spectroradiometer (MODIS) instrument on board the Aqua satellite. Circles indicate CTD stations (grey: SPACES/OASIS and open ocean stations during ASTRA-OMZ, black: equatorial stations during ASTRA-OMZ, red: coastal stations during ASTRA-OMZ). Numbers indicate station number.**


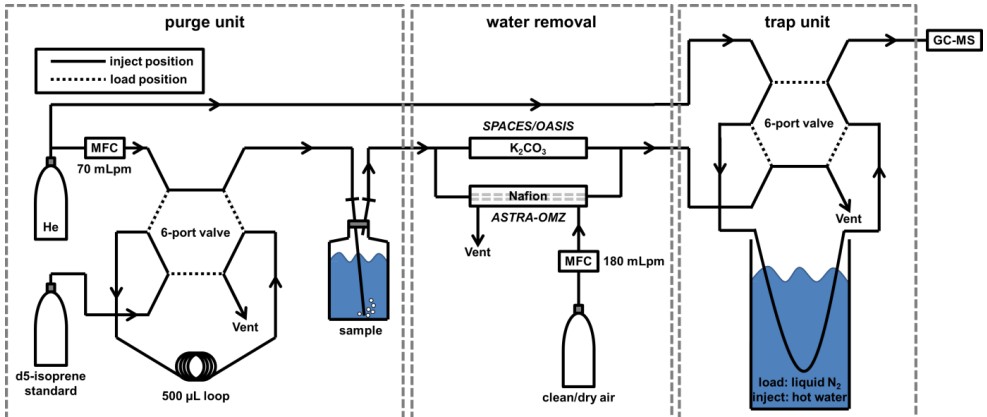

**Figure 2: Schematic overview of the analytical purge-and-trap-system, divided into three parts: purge unit (left), water removal (middle) and trap unit (right). He: helium, MFC: Mass flow controller, $K_2CO_3$: potassium carbonate,**
**GC-MS: gas chromatograph/mass spectrometer.**





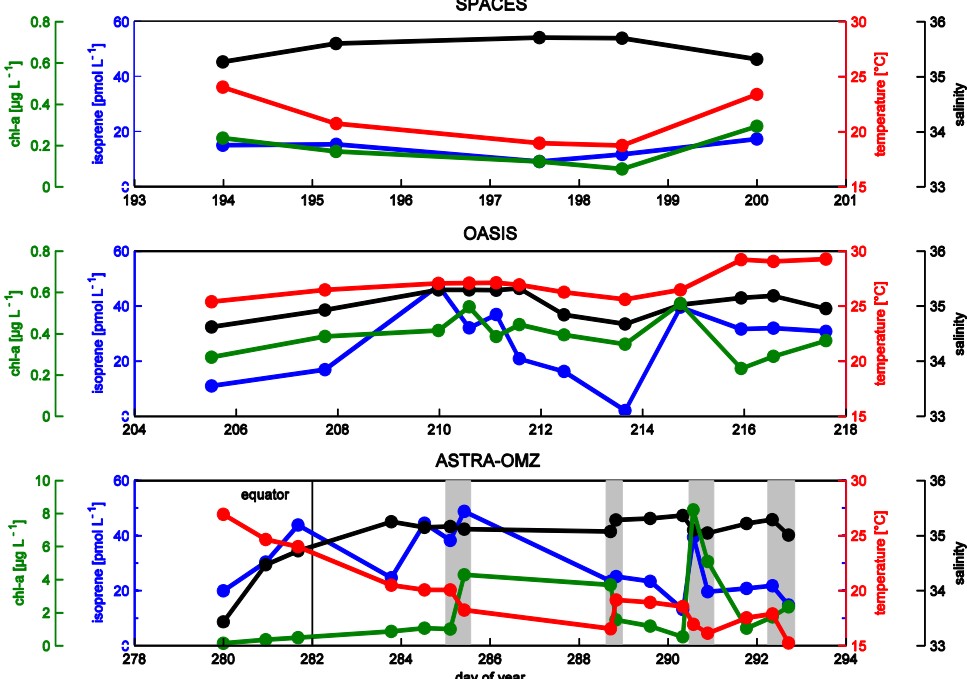

**Figure 3:** Mean salinity (black), isoprene concentration (blue), temperature (red), and chl-a concentration (green) in the MLD at each station during SPACES (upper panel), OASIS (middle panel), and ASTRA-OMZ (bottom panel). Grey rectangles highlight the 8 coastal stations during ASTRA-OMZ.




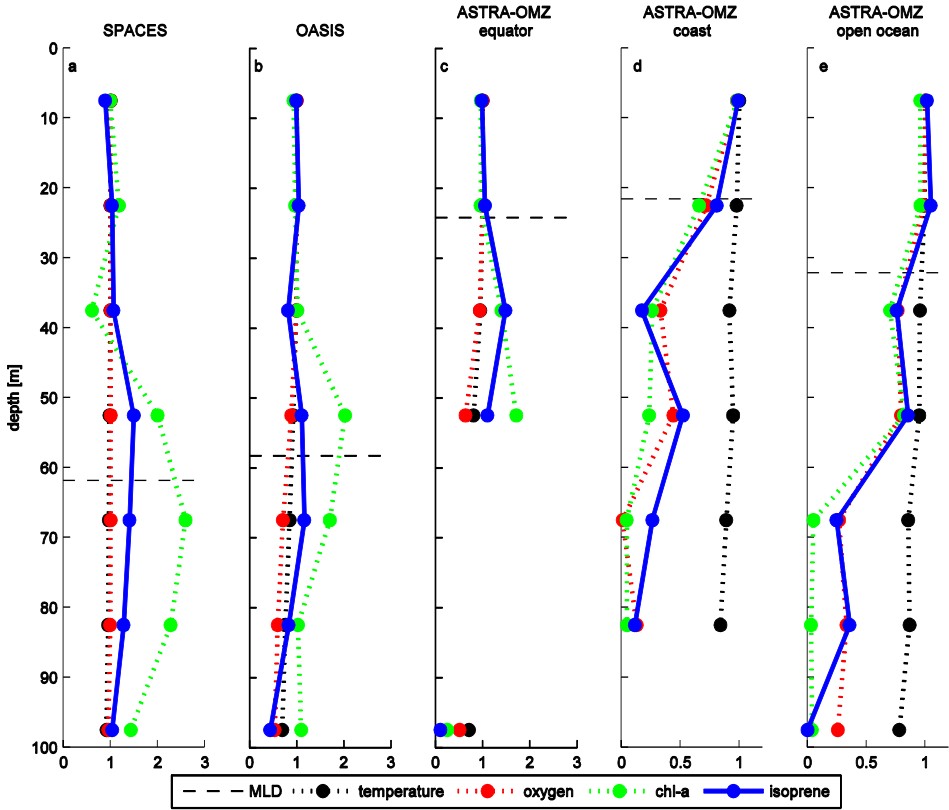

**Figure 4: Mean normalized depth profiles of temperature (black), oxygen (red), chl-a (green) and isoprene (blue) during (a) SPACES, (b) OASIS, and (c,d) ASTRA-OMZ (c: open ocean, d: coast). The black dashed line represents the mean MLD for each cruise.**






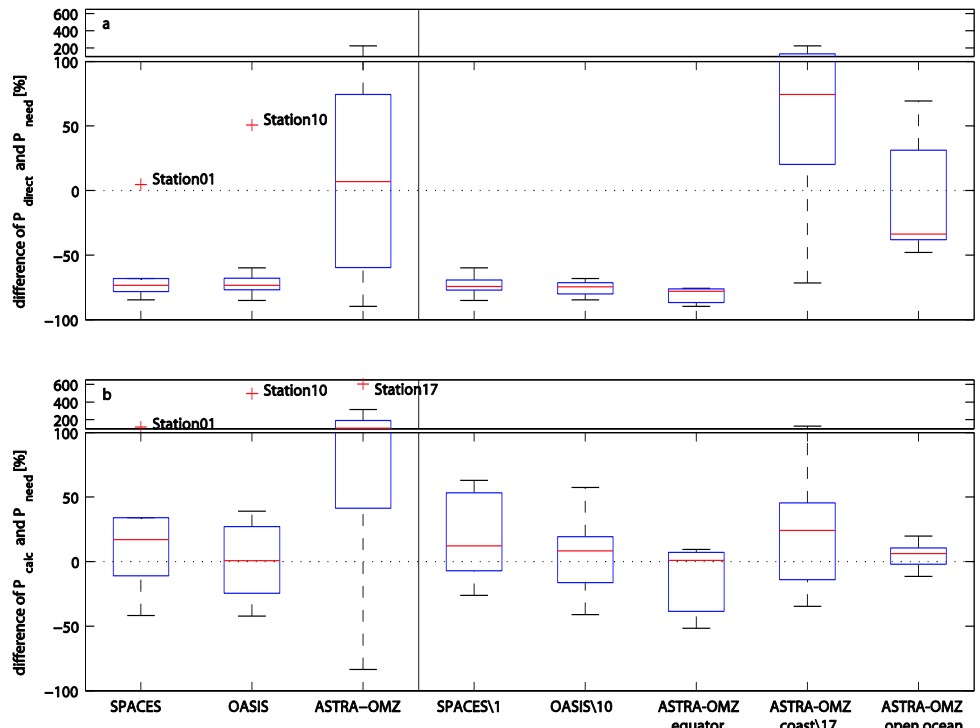

**Figure 5: Percent differences between (a) $P_{direct}$ and $P_{need}$ (($P_{direct}$-$P_{need}$)/$P_{need}$) and (b) $P_{calc}$ and $P_{need}$ (($P_{calc}$-$P_{need}$)/$P_{need}$)**
**for the different cruises / cruise regions. ASTRA-OMZ was split into three regions (equator, coast, open ocean). The red line represents the median, the boxes show the first to third quartile and the whiskers illustrate the highest and lowest values that are not outliers. The red plus signs represent outliers. The number indicated after \ denotes that a station that has been excluded from the analysis.**





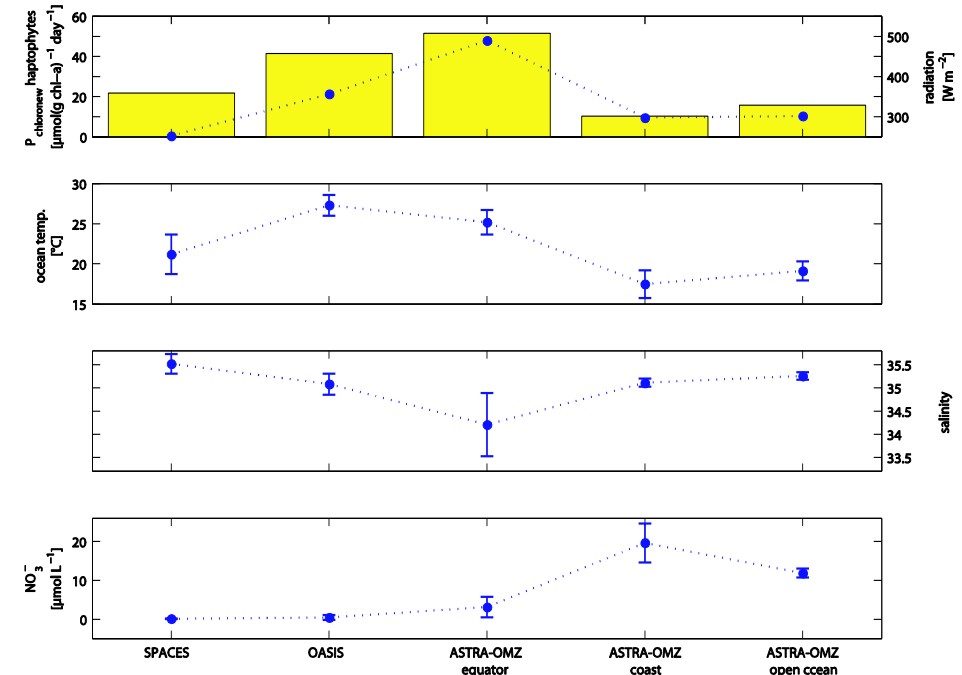


**Figure 6: Mean values for (a) calculated $P_{chloro}$ haptophytes (blue line) and global radiation (yellow bars), (b) ocean temperature, (c) salinity and (d) nitrate during SPACES/OASIS and ASTRA-OMZ (split into 3 different parts: equator, coast and open ocean).**






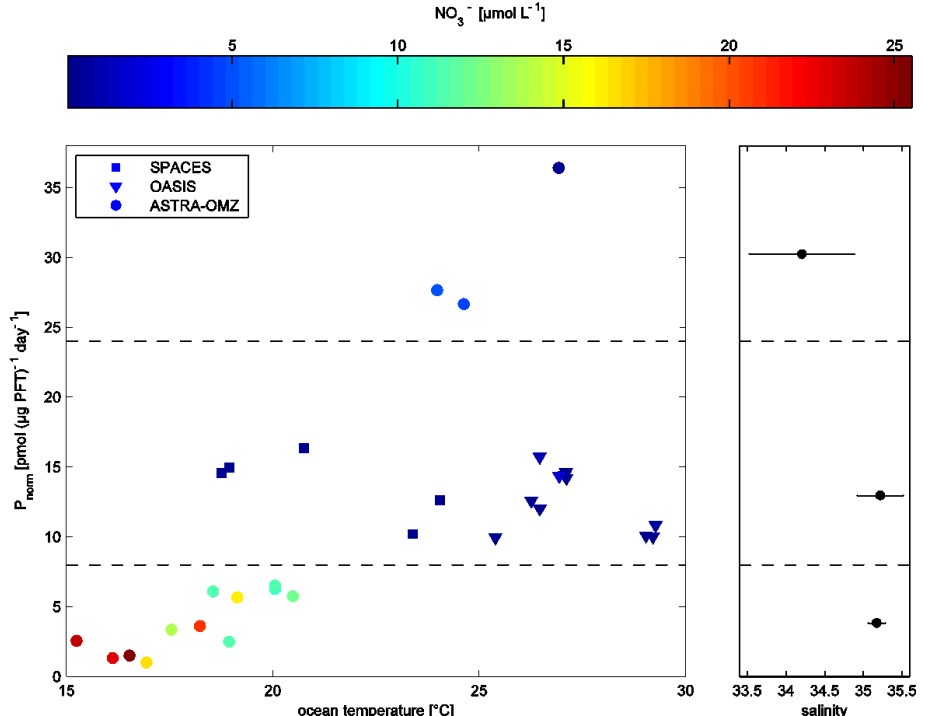

**Figure 7: Left panel:** Relationship between $P_{norm}$ in pmol (µg PFT)$^{-1}$ day$^{-1}$ and ocean temperature in °C during SPACES (squares), OASIS (triangles), and ASTRA-OMZ (circles) color-coded by NO$_3^-$ in µmol L$^{-1}$. **Right panel:** mean salinity of samples from left side plot in each box divided by dashed lines.






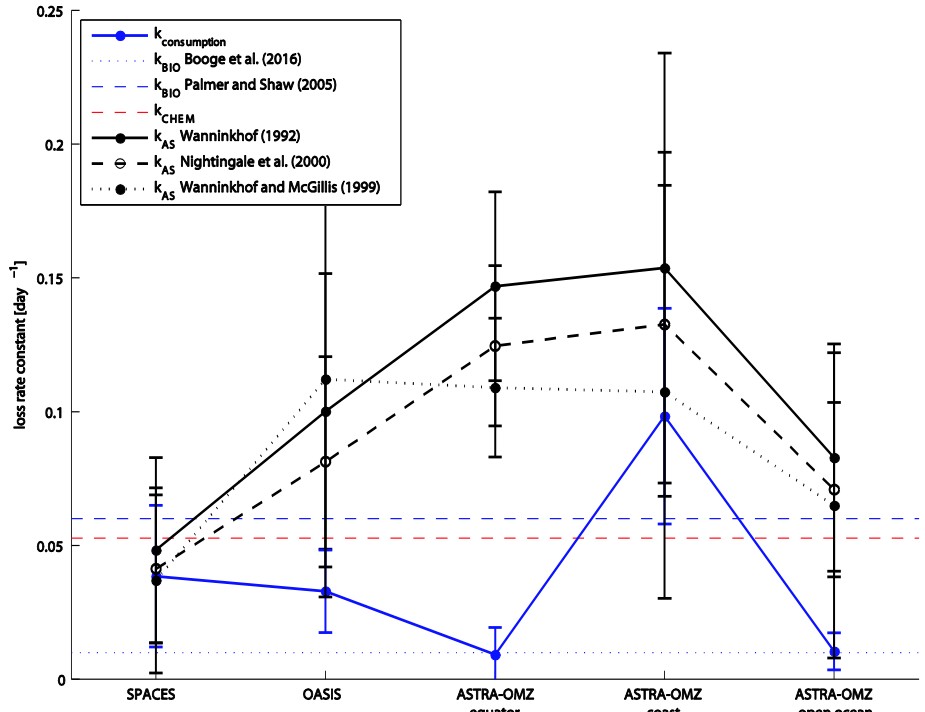

**Figure 8: Different mean loss rate constants during SPACES, OASIS und ASTRA-OMZ. Blue points: calculated loss rate ($k_{consumption}$), blue dotted line: $k_{BIO}$ from Booge et al. (2016), blue dashed line: $k_{BIO}$ from Palmer and Shaw (2005), red dashed line: $k_{CHEM}$, black points: calculated loss rate constants due to air-sea-gas exchange.**





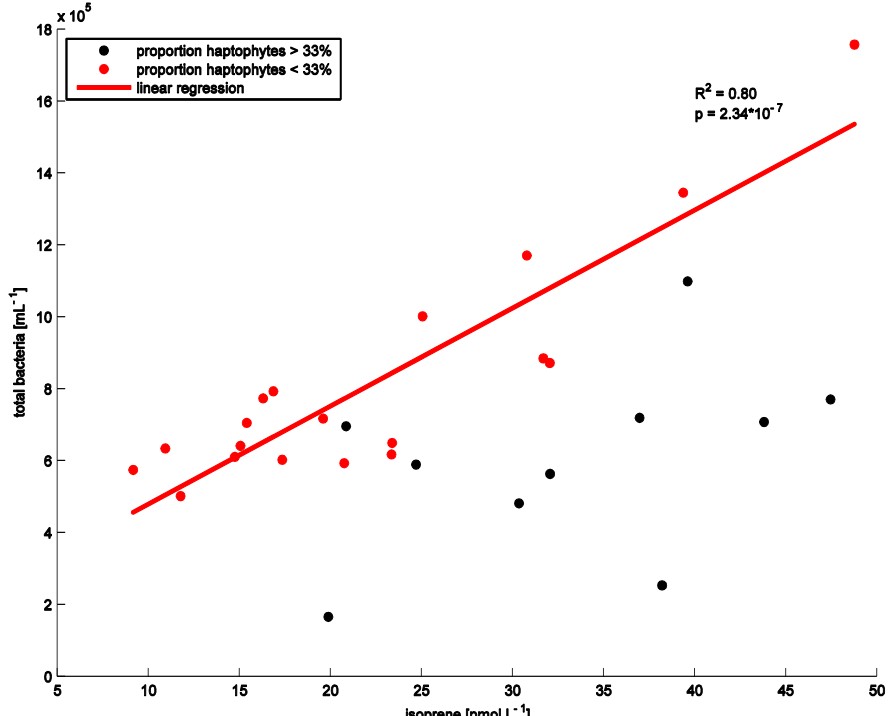

**Figure 9: Relationship between isoprene concentration [pmol L$^{-1}$] and total bacteria counts [mL$^{-1}$] during SPACES/OASIS and ASTRA-OMZ. Black and red points represent samples where the contribution of haptophytes to the total phytoplankton chl-a concentration is higher and lower than 33%, respectively. Linear regression (R²=0.80, p=2.34*10$^{-7}$) for red points only.**





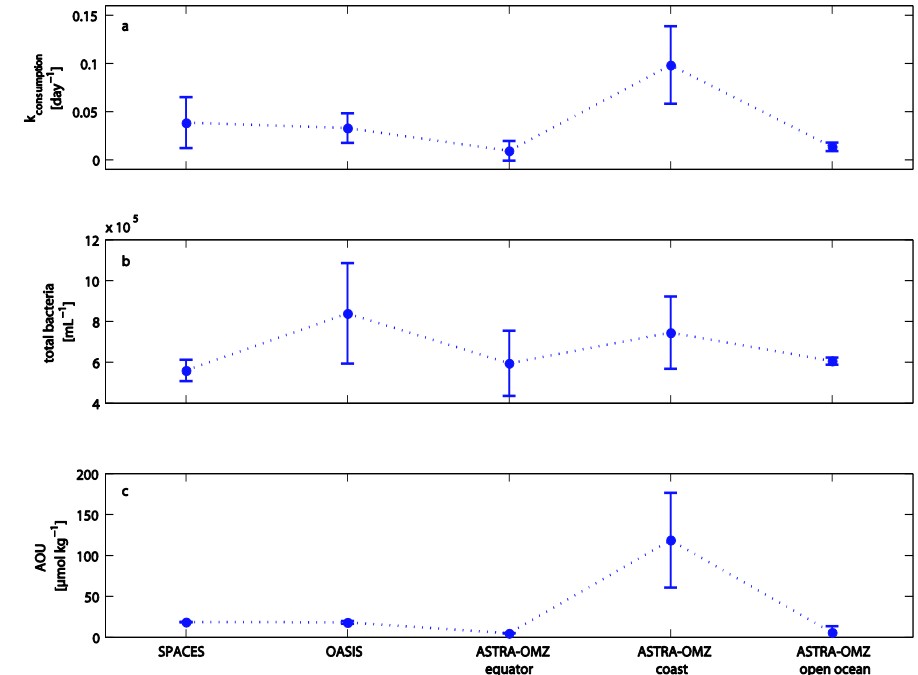


**Figure 10: Mean values for (a)** $k_{consumption}$ **[day$^{-1}$], (b) total bacteria counts [mL$^{-1}$] and (c) AOU [µmol L$^{-1}$] during SPACES/OASIS and ASTRA-OMZ (split into 3 different parts: equator, coast and open ocean).**