# Peer review of "Marine isoprene production and consumption in the mixed layer of the surface ocean – A field study over 2 oceanic regions"

_Biogeosciences, 2017_

## Referee Comment (RC1) · Anonymous Referee #1 · 21 Jul 2017

General Comments The authors report that there is a significant isoprene sink in the ocean, that needs to be accounted for to explain the observed concentrations of isoprene in the waters. In situ estimates of marine isoprene production is not made to the same extent of other biogenic hydrocarbons of global significance. Many of us still do not believe that marine isoprene is significant globally. It can be as little as 1 Tg per year if you accept conservative models, or some other more significant number, if you believe biologically meaningful assessment of empirical estimates. More studies such as the one by Booge et al will help us get closer to resolving this debate and help expand the research field of marine VOC-atmospheric interactions.

[Figure]

My biggest concern with this paper is the way in which the authors have assigned chlorophyll normalised isoprene emission rates to phytoplankton functional types *PFTs and also the emission factors derived from light response curves (tables 2 and 3, and the papers that are cited there). The authors themselves recognize clearly in the introduction and again in conclusion (L345-348, L480 onwards) that there are significant species-specific differences in isoprene emission capacities with respect to temperature (e.g. Exton et al. 2013) and light levels (e.g. Meskhidze et al 2015). Such studies are meaningful and important as individual studies. They may even provide a broad understanding of what a PFT does. There must be some caution while choosing species that are truly representative of a PFT while trying to derive an emission factor. Booge et al., have carefully left out species studied at subzero temperatures (which is a good thing as reflected in the table of Booge et al 2016 in ACPD (I have not read that paper fully). However, it is clear that they have included many species that are globally not relevant in terms of their abundance and those grown under different culture conditions. In those papers cited, cultures were grown at 16, 20 and 26 áȚŠC. SST is crucial for isoprene production. 10-degree increase can increase isoprene emission by 2 to 3 times over long term, and even higher levels over the short term in terrestrial ecosystems. E.g. In Table 3 of Exton et al (2013), they provide separate Pchloro for temperature and light response (irrespective of PFTs) and there are huge differences. Bonsang et al (2010) grew culture at a max light intensity of 100 umol/m2/s, Colomb et al (2008) did it at 250 umol/m2/s, Exton et al (2013), did measurements at 100 to 300 umol/m2/s. For all of these reasons I worry about the tenuous discussion on the Pchloro, and Pdirect presented in this paper.

Specific Comments L170 onwards and again L290 onwards: You say that haptophytes were the most dominant PFT in all three cruises (L330) and diatoms were dominant in coastal upwelling zone (figure s4). How do you explain fig s3, where haptophytes have very low emission response at light intensities <200 umol/m2/s, which is lower than that of diatoms. From your own figures (S1 and S2) light intensity below 10 m of the sea surface was less than 100 umol/m2/s. How can EF of haptophytes (L335)

be greater than that of diatoms at the working light intensities in the ocean? Why use single point light response curves (figure s3) for cryptophytes and dinoflagellates? What species were used to obtain those curves in figure s3? See figure 1 of Gantt et al (2009, ACP). They have a light response curve that is based on measurements made at 4 or 5 different light intensities for each PFT and responses are strikingly different to what you are proposing. Why wasn't their study considered in Table 2?

L185 and L288 onwards: The big difference between Pneeded and Pdirect is most likely due to the way you have calculated Pchloro, since Pdirect is largely dependent on EF (which is highly sensitive to temperature, light intensity, and species distribution). You rightly identify this as a potential reason (L300) but as highlighted earlier, the justification is difficult. In the equatorial region Pdirect is lower than Pneeded (figure 5) because of high SST and possibly also due to low emission factor you are assigning to cyanobacteria. The discrepancy in diatoms dominated coastal waters during ASTRA-OMZ is noteworthy. The spike in isoprene in site 14 and 15 correlates with diatom blooms in coastal upwelling zone. But, chlorophyll normalised emission suggests an overestimation of Pdirect in coastal sites. Isoprene is mixed quickly in MLD (as you rightly say in L265), hence no vertical trend above MLD. But, what about the relative contribution of phytoplankton below and above MLD to isoprene? Since the mixed layer is very shallow in coastal sites (figure 4d), is it possible that a large proportion of isoprene is locked below MLD? You do mention advective mixing in the thermocline being a slow process (L444). If you know phytoplankton abundances below and above MLD (likely also a function of plankton size), it is perhaps possible to understand this. Can this hold for the entire cruise, given that MLD generally was lower here compared to SPACES- OASIS? You also have a significant proportion of chlorophytes in these waters (figure S5) and they don't emit isoprene at high rates. What was their light response like?

L272-274: What you say in L280-284 is more appropriate than what you say here. Most of the previous studies have shown positive correlation between chl-a and isoprene

concentration (as Table 1 shows) in the oceans. The role of SST is also pretty well established.

L280-284: There is strong correlation between SST, chl-a, and isoprene concentration in cooler waters during both SPACES and ASTRA-OMZ cruises. (summarised in Table 1). Easy to see also in figures 3 and 7, but not mentioned. The relationship seems to breakdown at temperatures >25 deg C. Why? The discussion on relationships between chl-a, SST and isoprene is not satisfactory.

L379: "Higher temperatures caused a decrease in isoprene production rate [in diatoms]. ...If this temperature dependence can be transferred from diatoms also to haptophytes..." Yes. surely to Emiliania but perhaps to not all haptophytes. Please cite Heurtas et al. 2011 (Proc B) and a more recent meta-analysis from Chen, 2015 (J Phyt Res). However, I must point out that the discussion on cyanobacteria and Prochlorococcus is not satisfactory. Together they are 40% of the total biomass during SPACES-OASIS. They emit isoprene at high rates and considering how abundant they are, how tolerant they are even to temperatures >30 degC, they are really important to this discussion.

L463-465 and Figure 9: Assuming that bacterial consumption/degradation of isoprene does occur (L435, 454), you say that bacterial count correlates well with isoprene concentration in waters not dominated by haptophytes. I am not sure how anyone can conclude that high bacterial counts with increasing isoprene in waters translates to bacterial metabolism of isoprene. On the contrary, bacterial emission of isoprene is demonstrated (papers from the 1990s) and we don't know if some marine bacteria produce isoprene. Are there any specific reasons why bacterial counts are less when haptophytes are dominant? Do you mean bacterial populations thrive in waters not dominated by haptophytes? Is it that haptophytes are producing isoprene but there are other biogenic inhibitors that checks bacterial colonies in their vicinity? please explain My suggestion to you/ a clue: Haptophytes are the biggest consumers of bacteria in the ocean (please cite Unrein et al. 2013, ISME J). Now, please reassess figure 9.

L487-490: "The results show that the isoprene production is influenced by light, ocean temperature, and salinity, with an indication that the nutrient regime might exert some influence". The same point has been made more elaborately by others (please cite Loreto and Dani 2017, Trends Plant Sci), where PFTs, temperature, nutrients and their impact on isoprene is anticipated including the parallel you seem to draw between dimethylsulphoniopropionate and isoprene (L385).

Technical comments: L52-54: It is one thing finding extraordinary numbers and then it is quite another explaining how and why? Your own estimates are closer to what we know from other marine waters.

L170: "Isoprene production rates of different PFTs were determined in laboratory phytoplankton culture experiments (see Table 2 in Booge et al. (2016))". The measurements listed in the original table are also sourced from literature. Please state the same.

Figure 6: Missing letters a, b, c, d in subfigures

Figure S4: In a few sites, the category of others is really big. ?

―――――――――――――――――――――――

---

## Referee Comment (RC2) · Anonymous Referee #2 · 30 Jul 2017

General comments

This manuscript reports a new data set of isoprene depth profiles alongside supporting data from the Pacific and Indian Oceans, which is subsequently analysed for production and loss rates in the mixed layer. On the whole, the data presented in this work is a valuable addition to the existing global isoprene data set, along with the analysis of the results in a novel approach, with relevant supporting data to investigate suggested relationships, and fits into the scope of the journal.

A comparison with available literature parameterisations is made, with the valid conclusion that none are currently adequate for global predictions. To consolidate essentially

bottom-up and top-down production rates based on literature, the authors calculate a new field-based production rate, and subsequently suggest that a further adjustment from a significant and variable biological loss is needed to explain their isoprene observations. The analysis of the new data does not produce significant, quantitative correlations, but some interesting qualitative comparisons to several environmental variables appear to support the assignments to stress-related production and to losses to heterotrophic respiration.

The conclusions suggest investigation of different avenues which would add new insights into processes at various levels (semi-qualitative for heterotrophic respiration with large natural variability, quantitative for air-sea gas exchange losses) as well as repeating existing hypotheses supported by the new data analysis (environmental factors affect isoprene production).

Specific comments (major)

Line 113: Did you test for matrix effect/purge efficiency differences between MilliQ and seawater?

Line 177: Were detailed light intensities (and light cycle timings) available and comparable for all literature values? How did the authors account for potential effects of temperature variations (and growth stage) between studies?

Line 336-341/Table 3: Double-check literature values for Prochlorococcus and diatoms are correct (should exclude Arnold et al., 2009, as described in Hackenberg et al., 2017). The difference between diatom Pcalc and literature is rather large, but both are described as "low". Prochlorococcus are in fact within a similar low range, using Shaw et al. (2003) production rates.

Line 372: Are mean radiation values for ASTRA-OMZ equator, as opposed to the lower mean values described for open ocean and coastal regimes in the next sentence (Fig 6 suggests yes)? Also, the global radiation for those two is lower than SPACES, but
Pchloronew is higher for both, which is qualitatively consistent within ASTRA-OMZ, but not with the previous description across all cruises - this could perhaps be worded more clearly, e.g. line 373 "production rate was lower than around the equator".

Line 381: A caveat (transfer of dependence from diatoms to haptophytes) has already been noted by the authors, but it may also be worth considering that temperature effects may be just as variable as light effects between different species and hence also PFTs (cf. reference to Srikanta Dani, 2017, line 353).

Line 430: Would stations where a loss term was not needed not still represent part of the range of required potential additional loss terms, so that they should be included in the averages? Line 443: Both OASIS and ASTRA-OMZ open ocean kAS are 0.1 day-1, while the loss rates are 0.05 day-1 for SPACES and 0.15 day-1 for ASTRA-OMZ - why are SPACES and OASIS considered more comparable to kconsumption than the others?

Line 486: Has the effect of salinity been shown before? Could describe that stress (from light and temperature) has also been shown to be a factor.

Line 497: It is (almost?) impossible to exactly know all the different processes, as there are so many different factors and variations, e.g. just the number of phytoplankton and bacteria species and their exact distribution in the ocean at any one time. Our understanding of global marine isoprene cycling depends on a better knowledge of the involved systems and processes, but I hope that we can make significant progress even without exact knowledge... (The statement also suggests that knowing processes for PFTs in general may not be sufficient, as large variations within PFTs do occur - in contrast to the use of average rates in this manuscript.)

Line 495 etc: What is the authors' view on the relative importance of uncertainty due to variations within PFTs compared to air-sea gas exchange? The large variation for haptophytes, for example, is much larger than differences in kAS . As a result, could the suggested missing sink not also be explained at least partially by the presence of

a much lower-producing species of haptophytes?

Specific comments (clarifications/additions needed)

Line 56: Please also cite Moore and Wang 2006 and Hackenberg et al. 2017; both also show depth profiles.

Line 57/Table 1: The correlation shown in Kurihara et al. 2010 is for isoprene between 5 and 100 m depth, not only surface waters.

Line 100: Can you give more details for the vials used? (e.g. custom-made/manufacturer, how is the headspace achieved)

Line 139: Can you re-word " to relate... diagnostic pigments" to clarify the sentence? I can't follow what it means.

Line 140: Specify that [PFT] in the remaining text refers to the chl-a concs of each PFT. Lines 150-153: Can it be made clearer which steps were a separate step and which were a more detailed description of a previously mentioned step? Also, line 153-155: could clarify by deleting "last" and changing to "... profile, the Ctot and Zeu values from this last integration" (it was not immediately clear whether the last or second to last set of values was referred to). Line 152: How was determined which equation needed to be used?

Line 157-163: Is EdPAR(0-) in W m-2 before conversion to PARsurface ? If so, please explain the unit conversion more clearly. The text changes from using subsurface irradiation to surface irradiation without giving details of why these are equivalent. Also, why was the measurement used in those units if it was also available in umol m-2 s-1 (line 146)?

Line 163: Does EdPAR(0+) refer to surface irradiance as initially defined? If so, why is it used in a depth profile, while a correction is necessary for subsurface radiation EdPAR(0-)?

Lines 172 and 484 and Table 3: This suggests that Booge et al. 2016 contains new laboratory data; please specify that it is a collection of literature values, also in Table 3.

Line 181-187: This paragraph was slightly difficult to follow. Which depth does "each depth" refer to (isoprene sampling depth? 1-m bins?)? If pigment data and hence [PFT] was only available at a variable, small number of depths within the MLD at each station, how does this affect Pdirect given that it is calculated as the "sum of all products", which presumably means at all measured depths? Would a sum of two depths not result in higher production than a single depth, if all depths display similar [PFT] and production rates? Please clarify the paragraphs on these calculations, including how they relate to the introduction to section 2.7 (one production rate per station vs. different numbers of depths used).

Line 198: Mean wind speed/temperature taken from satellite in situ or from 24h of shipboard observations (not at the same site as CTD)?

Line 305 etc: Please specify if these calculations (and any others in the manuscript) were performed only for MLD data. This is not always clear where results are referred to after the initial presentation of the profiles.

Line 425: Can you re-word "these cruises" to be more specific? OASIS is mentioned separately due to a higher kAS (Wanninkof and McGillis, 1999), so it can't mean all three cruises in this work?

Line 449-451: While the statement that rates should be evaluated in water (and possibly in seawater, due to matrix effects?) is valid, the singlet oxygen reaction rate in Palmer and Shaw (2005) is in fact for chloroform (from Monroe, 1981).

Line 464: Should this be "isoprene concentration is no longer correlated to bacteria abundance", rather than referring to the isoprene production rate?

Line 467: Please clarify "it is important to scale the loss" - why is it important/in order to do what?

Line 468: Caused by the presence of different bacteria or by differences in their ability to use isoprene (or both)?

Lines 473-475: This point has effectively been previously made in other studies. Environmental factors/stresses such as temperature and light are already known to influence biological activity, and that in turn is already known to influence isoprene production.

Line 489: Ideally, use a different word instead of "show" - the results support existing theories/knowledge that these influences exist (described just before this), as opposed to showing something new. The salinity and nutrient relationships specifically do appear to support the hypothesis of stress-related isoprene production.

Lines 499-502: What exactly do you mean by this? Do the parameterisations need to be assessed, i.e. are specific factors for isoprene needed? Generally agreed values are not even available for the most common gases studied. It is worth pointing out that the parameterisation chosen will affect each study, so that perhaps it is useful to present different results if possible/relevant in a study.

Line 502: Could "The evaluation [...] should be examined" be worded differently?

Line 694 (Table 1): bold/italic is defined, but what are the R2 values that are neither?

Fig 1: Why are not all station numbers shown? Where they are shown, it is often difficult to assign them to a particular dot. There also seem to be stations omitted or not visible? If they cannot be shown (same location as another one) or were not sampled (as suggested by Fig 3), please add this information to the caption. It may also be useful to add station numbers to Fig 3 to connect the two pieces of information.

Fig 5: Can you please show n in this figure for each set of data and add some details to the caption about the left vs. right part of the graph or refer to the main text (especially 5b) to clarify? Also, why are most of the whiskers for SPACES and OASIS in 5a different once the outliers have been excluded (other values should not be affected if one point

is removed)? (For 5b, the new calculations can explain the changed whiskers, but are only mentioned in the main text.)

Fig 6, 7, 8, 10: What do the error bars show? Error on measurement or standard deviation of the average? Please add this information to the caption.

Fig S2: Why was EdPAR(0+) calculated if there were also measurements available (binned data implies measured)?

Fig S3: Why are chlorophytes and cyanobacteria functions not shown (EFs are listed in Table 2)? Please add to plot or add reason to caption.

Technical comments

Line 49: Change to "the concentrations generally range", as the following sentence presents different concentrations.

Lines 76 and 454: reference should be Acuña Alvarez

Line 131: Use "Phytoplankton functional types..." as heading for consistency

Lines 133, 146 and 150: Change to "same stations as isoprene was sampled"; "sub-surface irradiation", to define EdPAR(0-); and to "...the total chl-a concentration inte-grated..."

Line 139/140: Replace "By that" with something like "This was used to derive..." or "The chl-a concs... were derived that way"

Line 143 etc: Can PAR stand for both photosynthetically active radiation and photosyn-thetic available radiation? The latter does not seem commonly used.

Line 163: EdPAR(0+) should have superscript and be in italics? (also in Fig S2?)

Line 167: Suggest changing to "...due to aÅň shallow mixed layer depth (MLD) resulting in only one..."

Line 254-256: Either the numbers or the description appears to be the wrong way

round; dividing the mean by the concentration at a certain depth would give >1 for a smaller specific concentration.

Lines 300, 318, 453: punctuation before "2)" is almost invisible; remove comma after "which"; add comma after halocarbons

Line 308/318: Is there a difference between >80% of "total PFTs" and "total phyto-plankton chl-a"? If not, this statement is only needed once.

Line 334, 357, 487: change "than" to "from"; "stations"; "in-field production rates"

Line 388: "more saline" or "higher salinity"

Line 441: Add "Here, [the loss rate constant...]" to start of the sentence to clarify.

Line 499: must be further assessed? Furthermore, air-sea [...] has to be assessed?

Line 504: evaluate "their" impact (of the isoprene concentrations - if this refers in fact to the evaluation of the processes, the sentence is not very clear and should be re-worded)

Line 507: A link to the database would be useful.

Lines 704 and 738: (Table 3 and Fig 5 captions): remove the first "that"

Fig 1: x-axis values partially obscured for OASIS/SPACES

Fig 4 and Line 252 / Fig 8 and Lines 417-434: A darker shade of green would be easier to see (Fig 4); dotted lines are quite faint and legend covers error bar (Fig 8). Legend and description duplicate the information needed, details are also not needed in main text. ASTRA-OMZ details are also already given above the plot; check (c/d/e) (Fig 4).

Fig 6 caption: Pchloronew , not Pchloro , according to main text?

Fig S1: y-axis is umol m-2 s-1, while caption refers to W m-2. If a conversion was made, please specify.

[Figure]

---

## Author Comment (AC1) · 28 Aug 2017

*General Comments The authors report that there is a significant isoprene sink in the ocean, that needs to be accounted for to explain the observed concentrations of isoprene in the waters. In situ estimates of marine isoprene production is not made to the same extent of other biogenic hydrocarbons of global significance. Many of us still do not believe that marine isoprene is significant globally. It can be as little as 1 Tg per year if you accept conservative models, or some other more significant number, if you believe biologically meaningful assessment of empirical estimates. More studies such as the one by Booge et al will help us get closer to resolving this debate and help expand the research field of marine VOC-atmospheric interactions.*

**We thank referee #1 for reviewing this manuscript and for providing helpful comments. We will address the comments in the following (bold). The lines refer to the originally uploaded manuscript.**

*My biggest concern with this paper is the way in which the authors have assigned chlorophyll normalised isoprene emission rates to phytoplankton functional types *PFTs and also the emission factors derived from light response curves (tables 2 and 3, and the papers that are cited there). The authors themselves recognize clearly in the introduction and again in conclusion (L345-348, L480 onwards) that there are significant species-specific differences in isoprene emission capacities with respect to temperature (e.g. Exton et al. 2013) and light levels (e.g. Meskhidze et al 2015). Such studies are meaningful and important as individual studies. They may even provide a broad understanding of what a PFT does. There must be some caution while choosing species that are truly representative of a PFT while trying to derive an emission factor. Booge et al., have carefully left out species studied at subzero temperatures (which is a good thing as reflected in the table of Booge et al 2016 in ACPD (I have not read that paper fully). However, it is clear that they have included many species that are globally not relevant in terms of their abundance and those grown under different culture conditions. In those papers cited, cultures were grown at 16, 20 and 26 ḁṯŠC. SST is crucial for isoprene production. 10-degree increase can increase isoprene emission by 2 to 3 times over long term, and even higher levels over the short term in terrestrial ecosystems. E.g. In Table 3 of Exton et al (2013), they provide separate Pchloro for temperature and light response (irrespective of PFTs) and there are huge differences. Bonsang et al (2010) grew culture at a max light intensity of 100 umol/m2/s, Colomb et al (2008) did it at 250 umol/m2/s, Exton et al (2013), did measurements at 100 to 300 umol/m2/s. For all of these reasons I worry about the tenuous discussion on the Pchloro, and Pdirect presented in this paper.*

- **We absolutely understand the concerns of referee #1. The production rates of different PFTs vary depending on temperature and light intensity, which is also stated in the manuscript (l.61, l.376), and, in every case there will surely remain uncertainty when averaging over different species of one PFT. In the following we would like to respond to the points stated by referee #1.**
**(1) Palmer and Shaw (2005) used bulk chl-a concentrations and a globally averaged production rate of 1.8 µmol (g chl-a)$^{-1}$ day$^{-1}$ in order to calculate the isoprene production in their model. In Booge et al. (2016) we could use actual isoprene field measurements in order to improve this model by a factor of ~10, using actual averaged PFT concentrations, rather than using bulk chl-a concentrations. The next step is now, as we tried in this paper,**

to include the light dependency of the different PFTs to test if these rates from laboratory tests are somehow suitable for calculating isoprene concentrations in the ocean.

(2) We set our focus on the light dependency due to the natural cycle of light, which is applicable to the entire ocean. Laboratory studies (e.g. Exton et al., 2013;Shaw et al., 2003) could show that almost all isoprene is produced during daytime with higher production during higher light levels. This is also applicable to a depth profile in the ocean with higher light levels at the surface and lower light levels with depth in the mixed layer, if we assume the temperature constant.

(3) You correctly mentioned the different temperatures at which the laboratory studies were carried out, and referring to Table 3 of Exton et al. (2013), that the temperature, averaged over all different PFTs, has an influence on the isoprene production rate. But as these production rates are chl-a normalized rates it is worth to look at the rates dependent on chl-a, which is shown in Figure 3 in Exton et al. (2013):

[Figure]

Fig. 3. The relationship between Chl $a$ concentration and isoprene production rate in laboratory phytoplankton cultures grown at three different temperatures, showing regressions for (A) strains grown at $-1°C$ (filled circles), (B) strains grown at $16°C$ (open circles), and (C) strains grown at $26°C$ (filled triangles; regression equation values are shown in Table 3). Also shown is an overall regression for all strains at all temperatures ("pooled"), and the SST-independent relationship used in previous global models of marine isoprene identified by Shaw et al. (2003; 0.13 $\mu$mol isoprene [g Chl $a]^{-1}$ $h^{-1}$). Outlying results for *Dunaliella tertiolecta* are omitted from the regression equations displayed here.

During our studies the chl-a concentration ranged from 0.1 up to 8 µg L$^{-1}$, which is at the very low end of the chl-a concentration range shown in Exton et al. (2013), (see where the black triangles (experiments at 26°C) and open circles (experiments at 16°C) are overlapping each other (note: their x-axis unit is g L$^{-1}$)). The highest measured isoprene production rates were obtained at 26°C (top black triangles) which were in a chl-a regime that does not represent our study areas.

(4) By including the influence of light intensities only in this study, we were able to examine the temperature influence independently. We had the opportunity to corroborate the temperature-dependence found during laboratory studies directly in the field, since we did not include it from the beginning of our analysis.

*Specific Comments L170 onwards and again L290 onwards: You say that haptophytes were the most dominant PFT in all three cruises (L330) and diatoms were dominant in coastal upwelling zone (figure s4). How do you explain fig s3, where haptophytes have very low emission response at light intensities <200 umol/m2/s, which is lower than that of diatoms. From your own figures (S1 and S2) light intensity below 10 m of the sea surface was less than 100 umol/m2/s. How can EF of haptophytes (L335) be greater than that of diatoms at the working light intensities in the ocean? Why use single point light response curves (figure s3) for cryptophytes and dinoflagellates? What species were used to obtain those curves in figure s3? See figure 1 of Gantt et al (2009, ACP). They have a light response curve that is based on measurements made at 4 or 5 different light intensities for each PFT and responses are strikingly different to what you are proposing. Why wasn't their study considered in Table 2?*

- **We agree that the isoprene production rates of haptophytes at 45 $\mu$mol m$^{-2}$ s$^{-1}$ and 75 $\mu$mol m$^{-2}$ s$^{-1}$ (data from Shaw et al. (2003) and Bonsang et al. (2010)) are lower than the production rates for diatoms at the same light levels (data also from Shaw et al. (2003) and Bonsang et al. (2010)). At higher light levels (300 $\mu$mol m$^{-2}$ s$^{-1}$) the production rate of haptophytes is higher than for diatoms. In order to calculate the emission factor (EF) of each PFT, we applied a log squared relationship (following the approach of Gantt et al. (2009)). Therefore, in comparison to the individual measurements, the log squared curve is overestimating the production rate at lower light levels, but also underestimation the production rate at higher light levels. However, it is the best fit for all three data. The same is true for every PFT when applying the log squared fit. Even though this fit is associated with uncertainties depending on the individual data, the isoprene production rate for each PFT is still an average value of all investigated species within one PFT (as applied in Booge et al. (2016)), but now has a light dependency implemented, with significant influence on production rates.**

  **To caution the reader, that there are uncertainties using a log squared fit, we added a sentence to line 300: "….2) uncertainty of using a light dependent log squared fit. Measurements from different laboratory studies used different species within one group of PFTs. All species within one PFT group were combined to produce a light dependent isoprene production rate (Figure S3), although the isoprene production variability of different species within one PFT group is quite high. This will certainly influence P$_{direct}$, but cannot explain the 70% difference between P$_{direct}$ and P$_{need}$ measured at SPACES/OASIS and ASTRA-OMZ (equator) (Figure 5);"**

  **Figure S1 actually can lead to the conclusion that the light intensity below 10 m of the surface was less than 200 $\mu$mol m$^{-2}$ s$^{-1}$, but this is not true for most of the data. This data shown is an example from the SPACES cruise at approximately 25°S, which was the cruise operating in the highest latitudes of all three cruises. The mean light intensity was higher for all other data at lower latitudes near the equator (shown in updated Figure S1). The depth profiles of the stations chosen in Figure S2 from ASTRA-OMZ were performed during sunrise and sunset, resulting in lower light intensities.**

  **We understand that this figure might lead to confusion as referee #1 stated, therefore, we changed figure S1 as follows: Instead of using one single day as an example, we calculated the total mean of all cruises of the hourly radiation measurements from the ship (Figure**

S1a) and a total mean calculated PAR over the course of the day, depending on depth (Figure S1b). We also included the mean MLD of each cruise for a better understanding of the light levels present when sampling different depths.

We followed the approach of Gantt et al. (2009) to determine our light dependent production rates in the original publication (and continue to do so here). Only one production rate was available in the literature for cryptophytes, dinoflagellates, and *Prochlorococcus.* For haptophytes and diatoms, we used 3 and 6 light dependent isoprene production rates, respectively, and assuming a similar log squared dependence, as Gantt et al. (2009) did for *Prochlorococcus* and *Synecchococcus* in their study.

We added a footnote to Table 2 stating that the specific species of tested PFTs can be found in the cited literature.

We did not add the laboratory derived production rates of Gantt et al. (2009) to our calculation because they only provide the EF and not the actual rates. Moreover, we do n ot understand how Gantt et al. (2009) calculated their emission rates (y-axis, Figure 1 in Gantt et al. (2009)). It seems they calculated the EF of *Prochlorococcus* by converting an isoprene production rate of 1.5 µmol (g chl-a)$^{-1}$ day$^{-1}$ derived by Shaw et al. (2003) at 90 µmol m$^{-2}$ s$^{-1}$ to an hourly rate, and applied a similar log squared fit  as observed for diatoms and coccolithophores. Shaw et al. (2003) used a 14 hours light cycle resulting in an hourly production rate of 0.11 µmol (g chl-a)$^{-1}$. When using their fit, you should expect a production rate of 0.11 µmol (g chl-a)$^{-1}$ h$^{-1}$, when using a light intensity of 90 µmol m$^{-2}$ s$^{-1}$. However, according to the figure 1 in Gantt et al. (2009), a production rate of 0.7 µmol (g chl-a)$^{-1}$ h$^{-1}$ is obtained at a light level of 90 µmol m$^{-2}$ s$^{-1}$. Gantt et al. (2009) report similarly high rates for al measured PFTs in their study. As an example, their measured isoprene production rates for diatoms (see blue line, Fig. 1 in Gantt et al. (2009)) are in the range of 1.3 and 1.8  µmol (g chl-a)$^{-1}$ h$^{-1}$ at light levels of about 350 and 750  µmol m$^{-2}$ s$^{-1}$, respectively. In comparison, the isoprene production rates from literature values we used for diatoms at light levels of 300-900 µmol m$^{-2}$ s$^{-1}$ were in the range of ~0.22 µmol (g chl-a)$^{-1}$ h$^{-1}$ (see Figure S3). Due to these big discrepancies, the EF of Gantt et al. (2009) is also strikingly higher than ours. We do not know how to resolve these differences, however we checked our calculations and cannot find an error. Therefore, we only used the approach, but not the data from Gantt et al. (2009) in our study.

*L185 and L288 onwards: The big difference between Pneeded and Pdirect is most likely due to the way you have calculated Pchloro, since Pdirect is largely dependent on EF (which is highly sensitive to temperature, light intensity, and species distribution). You rightly identify this as a potential reason (L300) but as highlighted earlier, the justification is difficult. In the equatorial region Pdirect is lower than Pneeded (figure 5) because of high SST and possibly also due to low emission factor you are assigning to cyanobacteria. The discrepancy in diatoms dominated coastal waters during ASTRAOMZ is noteworthy. The spike in isoprene in site 14 and 15 correlates with diatom blooms in coastal upwelling zone. But, chlorophyll normalised emission suggests an overestimation of Pdirect in coastal sites. Isoprene is mixed quickly in MLD (as you rightly say in L265), hence no vertical trend above MLD. But, what about the relative contribution of phytoplankton below and above MLD to isoprene? Since the mixed layer is very shallow in coastal sites (figure 4d), is it possible that a large proportion of isoprene is locked below MLD? You do mention advective mixing in the thermocline being a slow process (L444). If you know phytoplankton abundances below and above MLD (likely also a function of plankton size), it is perhaps possible to understand this. Can this hold for the entire cruise, given*

*that MLD generally was lower here compared to SPACES- OASIS? You also have a significant proportion of chlorophytes in these waters (figure S5) and they don't emit isoprene at high rates. What was their light response like?*

- **Yes, the referee is absolutely right, in equatorial regions $P_{direct}$ is lower than $P_{need}$ due to high SST. The temperature dependence of isoprene production rates is not included in the calculation of $P_{direct}$. As stated in our first comment (point (4)), this was exactly what we wanted to test, if the temperature dependence can be seen in field studies.**

  **Also, the initial production rate of cyanobacteria might be a reason that $P_{direct}$ is significantly smaller than $P_{need}$. This is what we could prove, when using our data to calculate new production rates ($P_{chloronew}$) using the multiple linear regression. As shown in Table 3, the new $P_{chloronew}$ values for cyanobacteria are 2 to 7 times higher for OASIS and SPACES, respectively.**

  **Yes, the referee is right, in contrast to the equatorial and open ocean regions, $P_{direct}$ is overestimated compared to $P_{need}$. In these coastal upwelling areas, diatom concentrations are highly elevated and are accounting for ~80% of all PFTs. We attributed this either to a missing sink in this upwelling area or to incorrect literature derived $P_{chloro}$ values of diatoms (lines 293-304). The newly calculated $P_{chloronew}$ values for diatoms show that only a production rate of 0.5-0.6 instead of 2.5 µmol (g chl-a)$^{-1}$ day$^{-1}$ (Table 3) is needed to be in better agreement with $P_{need}$ (Figure 5b).**

  **Advective mixing in the thermocline is a very slow process. If there is a strong concentration gradient with high concentrations of isoprene slightly below the mixed layer, this process might contribute to some point to these concentrations in the MLD. We checked the individual stations but concentrations are not higher below the MLD. This can also be seen in Figure 4, looking at the mean depth profiles.**

  **Light responses of all PFTs are shown in updated Figure S3. Thank you for pointing out this missing information.**

*L272-274: What you say in L280-284 is more appropriate than what you say here. Most of the previous studies have shown positive correlation between chl-a and isoprene concentration (as Table 1 shows) in the oceans. The role of SST is also pretty well established.*

- **Yes, again we agree with the referee that many previous studies could show a positive correlation between chl-a and isoprene. The trend is clear: the more chl-a, the higher the isoprene concentration; the higher the SST, the higher the isoprene concentration (to some extent). However, we wanted to point out that, despite this large scale trend, the relationships between chl-a (or SST) and isoprene for each study or subset of a study are not consistent (i.e. no unique regression equation which can adequately describe the correlation between chl-a and isoprene globally, shown in Figure A). For clarification we changed the sentence in line 272 to: "…it can be seen that, even if the correlations for most of the datasets are significant, there is no globally unique regression factor to adequately describe the relationship between chl-a (and SST) and isoprene."**

[Figure]

**Figure A: Visualization of all significant regression equations of relationships between isoprene and chl-a in different oceanic regions (data taken from Table 1 in the manuscript). Dotted line: SST<20°C, dashed line: SST>20°C, solid line: no SST bin.**

*L280-284: There is strong correlation between SST, chl-a, and isoprene concentration in cooler waters during both SPACES and ASTRA-OMZ cruises. (summarised in Table 1). Easy to see also in figures 3 and 7, but not mentioned. The relationship seems to breakdown at temperatures >25 deg C. Why? The discussion on relationships between chl-a, SST and isoprene is not satisfactory.*

- **The lowest SST measured during SPACES was higher than 18°C, meaning that data from SPACES is not included in the regression equation with the strong correlation ($R^2$=0.89, Table 1) which we think you are referring to. We mention the correlation of $P_{norm}$ and ocean temperature during ASTRA-OMZ (shown in Figure 7) in lines 401 and 402: "…the $P_{norm}$ values were lower (< 8 pmol $(\mu g\ PFT)^{-1}\ day^{-1}$) correlating with lower ocean temperatures." However this is a correlation between isoprene and $P_{norm}$ and not chl-a. If we look at temperatures lower than 25°C, Figure 3 might suggest, that there is a strong correlation (especially for SPACES) between chl-a, SST and isoprene but in fact the correlation coefficient of $R^2$=0.42 is not as high as one would expect from looking at the figure. Using the data from all 3 cruises the correlation is significant for both cases, >25°C and <25°C, but the correlation using temperatures <25°C ($R^2$=0.37) is not significantly higher than using temperatures >25°C ($R^2$=0.32). Therefore, a possible breakdown of a suggested relationship between chl-a, SST, and isoprene at temperatures >25°C is mathematically not proven. A breakdown of isoprene production rates of diatoms (and haptophytes) at temperatures >26°C is discussed in paragraph 3.4, lines 377-382.**

*L379: "Higher temperatures caused a decrease in isoprene production rate [in diatoms]. …If this temperature dependence can be transferred from diatoms also to haptophytes…" Yes. surely to Emiliania but perhaps to not all haptophytes. Please cite Heurtas et al. 2011 (Proc B) and a more recent meta-analysis from Chen, 2015 (J Phyt Res). However, I must point out that the discussion on cyanobacteria and Prochlorococcus is not satisfactory. Together they are 40% of the total biomass*

*during SPACES-OASIS. They emit isoprene at high rates and considering how abundant they are, how tolerant they are even to temperatures >30 degC, they are really important to this discussion.*

- **We added the references in the updated sentence starting in line 380: "Increasing ocean temperatures influence the growth rate of phytoplankton generally, but also differently within a group of PFTs. For haptophytes, Huertas et al. (2011) show that two strains of *Emiliania huxleyi* were not tolerant to a temperature increase from 22°C to 30°C, whereas *Isochrysis galbana* could adapt to the increased temperature. In general, the optimal growth rate temperature decreases with higher latitude (Chen, 2015), but the link between growth rate of phytoplankton and isoprene production rate is still not known. Assuming this temperature dependence can be transferred…"**
**We concentrated on the discussion of haptophytes, because this was the only PFT of the three most abundant PFTS that recurred during all three campaigns. Nonetheless, we can see the referee's point that we have to add the results for cyanobacteria and *Prochlorococcus* to the discussion. Therefore, we added a paragraph starting at line 368: "*Prochlorococcus* was one of the three most abundant PFTs during SPACES and OASIS, but concentrations decrease to almost zero in the colder open ocean and upwelling regions of ASTRA-OMZ (Figure 1), which confirms the general knowledge that *Prochlorococcus* is absent at temperatures <15°C (Johnson et al., 2006). Our newly derived production rates confirm the actual laboratory derived rates, demonstrating *Prochlorococcus* as a minor contributor to isoprene concentration. However, *Prochlorococcus* is especially abundant at high ocean temperatures, where isoprene production rates from the other PFTs show evidence of decreasing. Cyanobacteria concentrations (excluding *Prochlorococcus*) were also related to temperature, but in contrast to *Prochlorococcus*, cyanobacteria were still abundant in colder waters during ASTRA-OMZ. The different derived isoprene productions rates for SPACES and OASIS might be related to the different mean ocean temperature and light levels during these cruises. During SPACES, with lower ocean temperatures and lower light levels, compared to OASIS, the production rate is higher. This relationship would confirm the findings of two independent laboratory studies of Bonsang et al. (2010) and Shaw et al. (2003). Bonsang et al. (2010) tested two species of cyanobacteria of at 20°C and found higher isoprene production rates than a different species tested by Shaw et al. (2003) at 23°C and even stronger light intensities. However, Exton et al. (2013) measured the same rate as Shaw et al. (2003) at 26°C for one species, but a 5-times higher production rate for another species at the same temperature. This leads to the conclusion that the production rate is not dependent on one environmental parameter and varies from species to species within the group of cyanobacteria."**

*L463-465 and Figure 9: Assuming that bacterial consumption/degradation of isoprene does occur (L435, 454), you say that bacterial count correlates well with isoprene concentration in waters not dominated by haptophytes. I am not sure how anyone can conclude that high bacterial counts with increasing isoprene in waters translates to bacterial metabolism of isoprene. On the contrary, bacterial emission of isoprene is demonstrated (papers from the 1990s) and we don't know if some marine bacteria produce isoprene. Are there any specific reasons why bacterial counts are less when haptophytes are dominant? Do you mean bacterial populations thrive in waters not dominated by haptophytes? Is it that haptophytes are producing isoprene but there are other biogenic inhibitors that checks bacterial colonies in their vicinity? please explain My suggestion to you/ a clue:*

*Haptophytes are the biggest consumers of bacteria in the ocean (please cite Unrein et al. 2013, ISME J). Now, please reassess figure 9.*

- **Figure 9 shows that at stations where isoprene concentrations are elevated, bacteria cell counts are elevated, too. We do not know if these two parameters are linked primarily to each other but we considered that there might be a correlation of bacterial cell counts and isoprene concentration due to the abundance of isoprene attracting bacteria to feed and thrive. If there is a lot of isoprene to eat (e.g. energy source), the bacteria abundance could increase, independent of any phytoplankton influence. This would support the findings from Acuña Alvarez et al. (2009) who showed that isoprene production by phytoplankton could facilitate the amount of hydrocarbon-degrading bacteria. However, due to the relatively high production rate of haptophytes in comparison to the rate of bacterial consumption of isoprene, we hypothesized that this correlation could not be seen anymore when haptophytes were dominant (>33%).**
  **We gratefully acknowledge the information that haptophytes are important grazers of bacteria. This helps to explain our results (Figure 9) in a more reasonable way. We added the explanation for Figure 9 starting at line 459: " This is a high isoprene production rate and we could assume higher isoprene concentrations with higher concentrations of haptophytes. This relationship, however, is not evident (data not shown), which may be attributable to other processes masking this relationship. Multiplying the chl-a normalized isoprene production rate of 17.9 µmol (g chl-$a$)$^{-1}$ day$^{-1}$ with the chl-a concentration of haptophytes results in a mean isoprene production rate of ~ 3 pmol L$^{-1}$ day$^{-1}$, which is about 4 times higher than the mean calculated loss rate due to bacterial degradation over all cruises (~ 0.8 pmol L$^{-1}$ day$^{-1}$).  This could hide the correlation of isoprene concentrations with bacteria when haptophytes are dominant (>33%). In addition, haptophytes themselves are suggested to be the main marine bacterial grazers, compared to other PFTs (Unrein et al., 2014). This leads to the hypothesis that, if there is a lot of isoprene that can be used (e.g. as energy source) by bacteria, also the bacteria abundance will increase, independent of any PFT. However, if the phytoplankton community is dominated (>33%) by haptophytes, the isoprene concentration is no longer correlated to the bacteria abundance, due to the grazing of bacteria by haptophytes (**Fehler! Verweisquelle konnte nicht gefunden werden.**, total bacteria cell counts of black points are lower than of the red points at similar isoprene concentrations)."**

*L487-490: "The results show that the isoprene production is influenced by light, ocean temperature, and salinity, with an indication that the nutrient regime might exert some influence". The same point has been made more elaborately by others (please cite Loreto and Dani 2017, Trends Plant Sci), where PFTs, temperature, nutrients and their impact on isoprene is anticipated including the parallel you seem to draw between dimethylsulphoniopropionate and isoprene (L385).*

- **We thank the referee for pointing out this additional reference. We would like to make sure, however, that within our manuscript we focus on the production rate of isoprene. We changed the sentence starting line at 488 and added a second statement focussing on the actual rate of isoprene production: "The results confirm findings from previous laboratory studies that the isoprene production is influenced by light and ocean temperature, due to stress, and nutrients, due to their effect on changing phytoplankton communities and their abundances (e.g. Dani and Loreto, 2017;Shaw et al., 2010).**

**Moreover, our data leads to the conclusion that isoprene production rates in the field, irrespective of phytoplankton communities and their abundance, are influenced by salinity and nutrient levels, which has never been shown before."**

*Technical comments: L52-54: It is one thing finding extraordinary numbers and then it is quite another explaining how and why? Your own estimates are closer to what we know from other marine waters.*

- **We absolutely agree. However, in the introduction section of the manuscript we tried to give an overview about the current knowledge/findings related to the biogeochemical cycling of isoprene. As there are not many oceanic isoprene studies published, it is worth to give an overview about the concentration range of marine isoprene concentration, which also includes those publications with extraordinary numbers. We do not try to explain why those numbers are so high, but rather just present them as published results.**

*L170: "Isoprene production rates of different PFTs were determined in laboratory phytoplankton culture experiments (see Table 2 in Booge et al. (2016))". The measurements listed in the original table are also sourced from literature. Please state the same.*

- **We changed the sentence to: "…were determined in laboratory phytoplankton culture experiments (see a collection of literature values: Table 2 in Booge et al. (2016)) and…"**

*Figure 6: Missing letters a, b, c, d in subfigures*

- **Done. Thank you for pointing that out.**

*Figure S4: In a few sites, the category of others is really big. ?*

- **We think that the referee is referring to Figure S5, not S4. Yes, we agree that the proportion of "others" of the total phytoplankton chl-a concentration e.g. at station 1 during ASTRA-OMZ is 50%. Firstly, this is an exception and, secondly, "others" consists of several PFTs (i.e. 5 different PFTs). We tested and found that using the whole community for the calculations does not lead to different results in production rate and, furthermore, in some cases, to highly unlikely production rates for the less abundant PFTs. Therefore, we would like to keep our evaluation with the most abundant 3 PFTs.**

**References**

Acuña Alvarez, L., Exton, D. A., Timmis, K. N., Suggett, D. J., and McGenity, T. J.: Characterization of marine isoprene-degrading communities, Environmental Microbiology, 11, 3280-3291, 10.1111/j.1462-2920.2009.02069.x, 2009.

Bonsang, B., Gros, V., Peeken, I., Yassaa, N., Bluhm, K., Zoellner, E., Sarda-Esteve, R., and Williams, J.: Isoprene emission from phytoplankton monocultures: the relationship with chlorophyll-a, cell volume and carbon content, Environmental Chemistry, 7, 554-563, 10.1071/En09156, 2010.

Booge, D., Marandino, C. A., Schlundt, C., Palmer, P. I., Schlundt, M., Atlas, E. L., Bracher, A., Saltzman, E. S., and Wallace, D. W. R.: Can simple models predict large-scale surface ocean isoprene concentrations?, Atmos. Chem. Phys., 16, 11807-11821, 10.5194/acp-16-11807-2016, 2016.

Dani, K. G. S., and Loreto, F.: Trade-Off Between Dimethyl Sulfide and Isoprene Emissions from Marine Phytoplankton, Trends in Plant Science, 22, 361-372, 10.1016/j.tplants.2017.01.006, 2017.

Exton, D. A., Suggett, D. J., McGenity, T. J., and Steinke, M.: Chlorophyll-normalized isoprene production in laboratory cultures of marine microalgae and implications for global models, Limnology and Oceanography, 58, 1301-1311, 2013.

Gantt, B., Meskhidze, N., and Kamykowski, D.: A new physically-based quantification of marine isoprene and primary organic aerosol emissions, Atmospheric Chemistry and Physics, 9, 4915-4927, 10.5194/acp-9-4915-2009, 2009.

Palmer, P. I., and Shaw, S. L.: Quantifying global marine isoprene fluxes using MODIS chlorophyll observations, Geophysical Research Letters, 32, 10.1029/2005gl022592, 2005.

Shaw, S. L., Chisholm, S. W., and Prinn, R. G.: Isoprene production by Prochlorococcus, a marine cyanobacterium, and other phytoplankton, Marine Chemistry, 80, 227-245, http://dx.doi.org/10.1016/S0304-4203(02)00101-9, 2003.

Shaw, S. L., Gantt, B., and Meskhidze, N.: Production and Emissions of Marine Isoprene and Monoterpenes: A Review, Advances in Meteorology, 10.1155/2010/408696, 2010.

Unrein, F., Gasol, J. M., Not, F., Forn, I., and Massana, R.: Mixotrophic haptophytes are key bacterial grazers in oligotrophic coastal waters, Isme Journal, 8, 164-176, 10.1038/ismej.2013.132, 2014.

---

## Author Comment (AC3) · 28 Aug 2017

*General comments*

*This manuscript reports a new data set of isoprene depth profiles alongside supporting data from the Pacific and Indian Oceans, which is subsequently analysed for production and loss rates in the mixed layer. On the whole, the data presented in this work is a valuable addition to the existing global isoprene data set, along with the analysis of the results in a novel approach, with relevant supporting data to investigate suggested relationships, and fits into the scope of the journal.*

*A comparison with available literature parameterisations is made, with the valid conclusion that none are currently adequate for global predictions. To consolidate essentially bottom-up and top-down production rates based on literature, the authors calculate a new field-based production rate, and subsequently suggest that a further adjustment from a significant and variable biological loss is needed to explain their isoprene observations. The analysis of the new data does not produce significant, quantitative correlations, but some interesting qualitative comparisons to several environmental variables appear to support the assignments to stress-related production and to losses to heterotrophic respiration.*

*The conclusions suggest investigation of different avenues which would add new insights into processes at various levels (semi-qualitative for heterotrophic respiration with large natural variability, quantitative for air-sea gas exchange losses) as well as repeating existing hypotheses supported by the new data analysis (environmental factors affect isoprene production).*

**We thank referee #2 for the helpful suggestions and comments. We will address the comments in the following (bold). The lines refer to the originally uploaded manuscript.**

*Specific comments (major)*

*Line 113: Did you test for matrix effect/purge efficiency differences between MilliQ and seawater?*

- **Yes, we did purge efficiency tests with seawater and MilliQ and can confirm that the purge time and purge flow rate we used are sufficient to remove total amount of dissolved isoprene from our samples.**

*Line 177: Were detailed light intensities (and light cycle timings) available and comparable for all literature values? How did the authors account for potential effects of temperature variations (and growth stage) between studies?*

- **All references for the values we used provided a detailed light intensity description, as well as a light cycle timing, which we used to convert daily rates into hourly rates or vice versa. Shaw et al. (2003) and Exton et al. (2013) used a 14 h light and 10 h dark-cycle, whereas Bonsang et al. (2010) used a 12 h light and 12 h dark-cycle. The phytoplankton cultures from the different studies were reported as being in exponential growth stage. The potential effect of temperature variations was not considered and is discussed in answer #1 in response to referee #1.**

*Line 336-341/Table 3: Double-check literature values for Prochlorococcus and diatoms are correct (should exclude Arnold et al., 2009, as described in Hackenberg et al., 2017). The difference between*

*diatom Pcalc and literature is rather large, but both are described as "low". Prochlorococcus are in fact within a similar low range, using Shaw et al. (2003) production rates.*

- **In fact, we did not use the isoprene production rate for *Prochlorococcus* from Arnold et al. (2009) in our calculations (see reference for *Prochlorococcus* in Table 2) but forgot to exclude this value for comparison in Table 3. We changed the value in Table 3 from 9.66 to 1.5 μmol (g chl-a)$^{-1}$ day$^{-1}$, which is in a good agreement with our field derived calculated isoprene production rates for SPACES and OASIS. Accordingly, we changed the sentence starting on line 336 to: "During SPACES/OASIS the P$_{chloronew}$ values of *Prochlorococcus* (both 0.5 μmol (g chl-a)$^{-1}$ day$^{-1}$) are slightly lower but in a good agreement with the mean literature value (1.5 μmol (g chl-a)$^{-1}$ day$^{-1}$, Table 3), whereas..."**
**The literature value for diatoms is also changed (in Table 3 and line 340) from 2.54 to 2.51 μmol (g chl-a)$^{-1}$ day$^{-1}$ by excluding Arnold et al. (2009) from average literature isoprene production rate of diatoms.**

*Line 372: Are mean radiation values for ASTRA-OMZ equator, as opposed to the lower mean values described for open ocean and coastal regimes in the next sentence (Fig 6 suggests yes)? Also, the global radiation for those two is lower than SPACES, but Pchloronew is higher for both, which is qualitatively consistent within ASTRA-OMZ, but not with the previous description across all cruises - this could perhaps be worded more clearly, e.g. line 373 "production rate was lower than around the equator".*

- **We changed the sentences to: "Highest mean values were measured during ASTRA-OMZ (at equator, ~508 W m$^{-2}$)...the isoprene production rate was lower than around the equator (mean global radiation decreased to ~310 W m$^{-2}$)."**

*Line 381: A caveat (transfer of dependence from diatoms to haptophytes) has already been noted by the authors, but it may also be worth considering that temperature effects may be just as variable as light effects between different species and hence also PFTs (cf. reference to Srikanta Dani, 2017, line 353).*

- **We added the following sentence at line 382 etc.: "Additionally, as mentioned before, the temperature, as well as the light dependence of isoprene production might vary between different species of haptophytes when comparing different ocean regimes."**

*Line 430: Would stations where a loss term was not needed not still represent part of the range of required potential additional loss terms, so that they should be included in the averages? Line 443: Both OASIS and ASTRA-OMZ open ocean kAS are 0.1 day-1, while the loss rates are 0.05 day-1 for SPACES and 0.15 day-1 for ASTRA-OMZ - why are SPACES and OASIS considered more comparable to kconsumption than the others?*

- **We assume that isoprene production by phytoplankton is the only source for isoprene in the water column. To date, we do not exactly know all different processes of isoprene production/consumption, so there could be other production and loss processes that are not included yet, but would balance each other out. We only used those stations where a loss was needed mathematically, in order to assess loss processes where we expected a large signal. We realize a more thorough assessment would need an iterative approach**

between sources and sinks. However, we focused here on getting a more basic understanding of the important loss processes in the field and we hope to investigate these loss processes in more detail in the future.

We thank the referee for pointing out this mistake in comparing k values and we changed the sentence starting at line 441 to: "…resulting in a lifetime of isoprene of only 10 days, which is comparable to the lifetime due to air sea gas exchange during ASTRA-OMZ (open ocean) and OASIS."

*Line 486: Has the effect of salinity been shown before? Could describe that stress (from light and temperature) has also been shown to be a factor.*

- **To our knowledge, the possible salinity stress of phytoplankton to produce isoprene has not been shown before. In addition we changed the sentence starting at line 488:" The results confirm findings from previous laboratory studies that the isoprene production is influenced by light and ocean temperature, due to stress, and nutrients, due to their effect on changing phytoplankton communities and their abundances (e.g. Dani and Loreto, 2017;Shaw et al., 2010). Moreover, our data leads to the conclusion that isoprene production rates in the field, irrespective of phytoplankton communities and their abundance, are influenced by salinity and nutrient levels, which has never been shown before."**

*Line 497: It is (almost?) impossible to exactly know all the different processes, as there are so many different factors and variations, e.g. just the number of phytoplankton and bacteria species and their exact distribution in the ocean at any one time. Our understanding of global marine isoprene cycling depends on a better knowledge of the involved systems and processes, but I hope that we can make significant progress even without exact knowledge... (The statement also suggests that knowing processes for PFTs in general may not be sufficient, as large variations within PFTs do occur – in contrast to the use of average rates in this manuscript.)*

- **We absolutely agree with this statement. However, in this study we could show in the field that, even using average rates, temperature has an effect on the production rates. This is also partially discussed in our answer #1 in response to referee #1. We often caution the reader about possible uncertainties, like large variations of isoprene production within the PFTs (e.g. lines 64, 346, 392), which we still are not able to implement correctly when modelling oceanic isoprene concentration. However, trends and qualitative correlations in the field can already be concluded (and support laboratory studies), without knowing every rate exactly, which will hopefully help to further understand global marine isoprene cycling.**

*Line 495 etc: What is the authors' view on the relative importance of uncertainty due to variations within PFTs compared to air-sea gas exchange? The large variation for haptophytes, for example, is much larger than differences in kAS . As a result, could the suggested missing sink not also be explained at least partially by the presence of a much lower-producing species of haptophytes?*

- **The calculated emission factor for haptophytes was derived from three different laboratory studies, using with four different species within the group of haptophytes cultivated under three different light levels and temperatures (Figure S3, Table 2). We think that is a good example for the variation of isoprene production under different environmental conditions**

within one group of PFT. The uncertainty of this emission factor (error of log squared fit) is ~56%, hence also for the $P_{direct}$ value. The uncertainty (standard deviation) of $k_{AS}$ using three different parameterizations is dependent on the wind speed and is 10-15% in a wind speed regime of 8-12 m s$^{-1}$ and can be 30% and higher at wind speed >15 m s$^{-1}$. Applying 15% uncertainty to the loss due to air-sea-gas-exchange (average: 2.88 pmol L$^{-1}$ day$^{-1}$) and 56% uncertainty the production by haptophytes (average: 0.89 pmol L$^{-1}$ day$^{-1}$) yields in an absolute error of 0.43 pmol L$^{-1}$ day$^{-1}$ and 0.50 pmol L$^{-1}$ day$^{-1}$ for the loss due to air-sea-gas-exchange and the production by haptophytes, respectively. As these two losses are both in the same range and following this approach and assuming that 56% uncertainty can be applied to all PFTs (and not only haptophytes) by using $P_{direct}$ it may be possible that the large variations within one PFT could account for the missing sink.

However, we computed $P_{need}$ values based on isoprene measurements, which allows us to disregard the uncertainties on $P_{direct}$. The resulting chl-a normalized isoprene production rates ($P_{chloronew}$) where highly variable among PFT (e.g. haptophytes) depending on the ocean region (Table 3). We hypothesize that these variations already reflect the influence of light, temperature, salinity, and nutrients. Hence, the uncertainty of the newly derived rates should be less than 56% (error of light dependent log squared fits from different laboratory studies using different temperatures and species), because these natural variations are already included. For this reason, we think that there has to be at least one missing sink, which accounts for the difference in $P_{calc}$ and $P_{need}$.

*Specific comments (clarifications/additions needed)*

*Line 56: Please also cite Moore and Wang 2006 and Hackenberg et al. 2017; both also show depth profiles.*

- **Thank you for pointing that out. We added Hackenberg et al. (2017), but not Moore and Wang (2006), as the sentence is about the comparison of chl-a and isoprene in a depth profile and they do not provide any chl-a data.**

*Line 57/Table 1: The correlation shown in Kurihara et al. 2010 is for isoprene between 5 and 100 m depth, not only surface waters.*

- **Sentence changed to: "…and furthermore, Broadgate et al. (1997) and Kurihara et al. (2010) show a direct correlation between isoprene and chl-a concentrations in surface waters and between 5 and 100 m depth, respectively."**

*Line 100: Can you give more details for the vials used? (e.g. custommade/ manufacturer, how is the headspace achieved)*

- **The sentence has been changed to: "10 mL of helium were pushed into each transparent glass vial (Chromatographie Handel Müller, Fridolfing, Germany) replacing the same amount of sea water and providing a headspace for the upcoming analysis."**

*Line 139: Can you re-word " to relate... diagnostic pigments" to clarify the sentence? I can't follow what it means.*

- **In the following we have explained in more detail this method. However, we think all this information can be easily obtained from the given citations in this text, so we would prefer to only slightly change the text (by adding only "to the concentration of monovinyl chlorophyll *a* concentration. The latter is an ubiquitous pigment in all PFT except *Prochlorococcus* sp. which contains divinyl chlorophyll *a* instead.." to the text) in order to keep the paper focused: "PFT was calculated using the diagnostic pigment analysis developed by Vidussi et al. (2001) and adapted in Uitz et al. (2006). This method uses specific phytoplankton pigments which are (mostly) common only in one specific PFT. These pigments are called marker or diagnostic pigments (DP) and the method relates for each measurement point the weighted sum of the concentration of seven, for each PFT representative DP to the concentration of monovinyl chlorophyll *a* concentration and by that PFT group specific coefficients are derived which enable to derive the PFT chlorophyll *a* (chl-a) concentration. The latter is an ubiquitous pigment in all PFT except *Prochlorococcus* sp. which contains divinyl chlorophyll *a* instead. In general, the chl-a is a valid proxy for the overall phytoplankton biomass. In the DP analysis as DP concentrations of fucoxanthin, peridinin, 19'hexanoyloxy-fucoxanthin, 19'butanoyloxy-fucoxanthin, alloxanthin, and chlorophyll *b* indicative for diatoms, dinoflagellates, haptophytes, chrysophytes, cryptophytes, cyanobacteria (excluding *Prochlorococcus* sp.), and chlorophytes, respectively, are used. With the DP analysis then finally the chl-a of these PFTs were derived. The chl-a concentration of *Prochlorococcus* sp. was directly derived from the concentration of divinyl chlorophyll *a*."**

*Line 140: Specify that [PFT] in the remaining text refers to the chl-a concs of each PFT.*

- **Every time we talk about the actual chl-a concentration of a PFT in the manuscript we now changed "PFT concentration" to "PFT chl-a concentration" to be more specific.**

*Lines 150-153: Can it be made clearer which steps were a separate step and which were a more detailed description of a previously mentioned step? Also, line 153-155: could clarify by deleting "last" and changing to "... profile, the Ctot and Zeu values from this last integration" (it was not immediately clear whether the last or second to last set of values was referred to). Line 152: How was determined which equation needed to be used?*

- **We have rephrased the whole paragraph and hope to have improved what exactly done. This should also clarify the two points mentioned below.**
  **For clarification which equation was used: You first apply Equation 2. When your $Z_{eu}$ is larger than 102 m you start again with the calculation using Equation 3 and taking the outcome of $Z_{eu}$ from there.**

*Line 157-163: Is EdPAR(0-) in W m-2 before conversion to PARsurface ? If so, please explain the unit conversion more clearly. The text changes from using subsurface irradiation to surface irradiation without giving details of why these are equivalent. Also, why was the measurement used in those units if it was also available in umol m-2 s-1 (line 146)?*

- **Please see above.**

*Line 163: Does EdPAR(0+) refer to surface irradiance as initially defined? If so, why is it used in a depth profile, while a correction is necessary for subsurface radiation EdPAR(0-)?*

- **Please see above**

*Lines 172 and 484 and Table 3: This suggests that Booge et al. 2016 contains new laboratory data; please specify that it is a collection of literature values, also in Table 3.*

- **Done. Thanks for pointing that out.**

*Line 181-187: This paragraph was slightly difficult to follow. Which depth does "each depth" refer to (isoprene sampling depth? 1-m bins?)? If pigment data and hence [PFT] was only available at a variable, small number of depths within the MLD at each station, how does this affect Pdirect given that it is calculated as the "sum of all products", which presumably means at all measured depths? Would a sum of two depths not result in higher production than a single depth, if all depths display similar [PFT] and production rates? Please clarify the paragraphs on these calculations, including how they relate to the introduction to section 2.7 (one production rate per station vs. different numbers of depths used).*

- **"Sum of all products" does not mean "sum over all depths". Following Equation 7 we multiplied for every sampled depth z the concentration of each PFT ($PFT_i$) with its (light-depth-dependent) $P_{chloro,i}$ value resulting in a production rate for $PFT_i$ at sampled depth z. To calculate the total isoprene production $P_{direct}$ at sampled depth z we summed up all individual production rates of all PFTs measured. In order to use only one production rate per station, we integrated the derived production rates of all measured depths z for each station over the total MLD. Scaling with the MLD gives us the total "mean" isoprene production within the mixed layer.**
  **We agree with referee #2 that these calculations are not described clearly in the text. For clarification, we changed the text, starting at line 180:"In order to calculate the isoprene production at each sampled depth (z) at each station, we used the scalar photosynthetic available radiation in the water column, PAR(z), (see section 2.6) as input for I, which was used with the respective, calculated EF of each PFT using Equation 6. The product was integrated over the course of the day, resulting in a $P_{chloro}$ value (µmol isoprene (g chl-a)$^{-1}$ day$^{-1}$) for each PFT and day depending on the depth in the water column (Figure S4). The light and depth dependent individual $P_{chloro,i}$ values of each PFT at the sampled depth z were multiplied with the corresponding, measured PFT concentration ($[PFT]_i$). The sum of all products gives the directly calculated isoprene production rate at each sampled depth z:**

$$P_{direct}(z) = \sum \left( P_{chloro_i} \times [PFT]_i \right). \tag{1}$$

  **Integrating over all measurements within the mixed layer and scaling with the MLD results in a "mean" direct isoprene production rate ($P_{direct}$) for each station."**

*Line 198: Mean wind speed/temperature taken from satellite in situ or from 24h of shipboard observations (not at the same site as CTD)?*

- **For clarification we changed the sentence to: "…, we used the mean wind speed and the mean sea surface temperature of the last 24 h of shipboard observations before taking…"**

*Line 305 etc: Please specify if these calculations (and any others in the manuscript) were performed only for MLD data. This is not always clear where results are referred to after the initial presentation of the profiles.*

- In paragraph 2.7, lines 166 etc. we state: "For all calculations made we came up with one production rate per station within the mixed layer. This was either due to…" We give this information right in the beginning of the method section to make clear that this is valid for the whole paper. For clarification we added this information again at line 305: "Therefore, we calculated new individual chl-a normalized production rates of each PFT ($P_{chloronew}$) within the MLD."

*Line 425: Can you re-word "these cruises" to be more specific? OASIS is mentioned separately due to a higher kAS (Wanninkof and McGillis, 1999), so it can't mean all three cruises in this work?*

- We changed the sentence to: "However, during SPACES and ASTRA-OMZ the wind speed was…"

*Line 449-451: While the statement that rates should be evaluated in water (and possibly in seawater, due to matrix effects?) is valid, the singlet oxygen reaction rate in Palmer and Shaw (2005) is in fact for chloroform (from Monroe, 1981).*

- Correct, we changed the sentence to: "It must be noted that the loss rate due to the reaction with OH is a gas phase reaction rate (Atkinson et al., 2004) and the used rate for reaction with singlet oxygen derives from measurements in chloroform (Monroe, 1981), meaning that these rates might not be suitable for isoprene reactions in the water phase."

*Line 464: Should this be "isoprene concentration is no longer correlated to bacteria abundance", rather than referring to the isoprene production rate?*

- Yes, we changed the sentence to: "…, the isoprene production rate is much higher than the degradation rate by bacteria and, therefore, the isoprene concentration is no longer correlated to the bacteria abundance."

*Line 467: Please clarify "it is important to scale the loss" - why is it important/in order to do what?*

- The loss rate constant of bacterial degradation is variable looking at the different regions (cruises). This means that this loss is not just a static number and therefore is dependent on something, such as environmental parameters or bacterial cell counts. For clarification, we changed the sentence starting in line 465: "Due to the different loss rate constants of bacterial degradation […] in the different regions it is important to identify their dependence on environmental parameters. "

*Line 468: Caused by the presence of different bacteria or by differences in their ability to use isoprene (or both)?*

- For clarification we changed the sentence to: "…, which may be caused by different heterotrophic bacteria, each with a different ability to use isoprene as an energy source."

*Lines 473-475: This point has effectively been previously made in other studies. Environmental factors/stresses such as temperature and light are already known to influence biological activity, and that in turn is already known to influence isoprene production.*

- Yes, the referee is absolutely right, it is known that environmental factors influence the isoprene production. The point we wanted to make is that the trend of higher loss rate

constant and higher AOU values might be a hint that also isoprene loss/consumption is actually influenced by biological activity and not only by air sea gas exchange or chemical loss.

*Line 489: Ideally, use a different word instead of "show" - the results support existing theories/knowledge that these influences exist (described just before this), as opposed to showing something new. The salinity and nutrient relationships specifically do appear to support the hypothesis of stress-related isoprene production.*

- **Changed to "The results confirm findings from previous studies…".**

*Lines 499-502: What exactly do you mean by this? Do the parameterisations need to be assessed, i.e. are specific factors for isoprene needed? Generally agreed values are not even available for the most common gases studied. It is worth pointing out that the parameterisation chosen will affect each study, so that perhaps it is useful to present different results if possible/relevant in a study.*

- **As isoprene is a very insoluble gas, like $CO_2$, we think the existing parameterisations are applicable to isoprene. We wanted to point out that there are different commonly used wind speed based k-parameterisations (i.e. Nightingale et al. (2000) or Wanninkhof and McGillis (1999)), which lead to different emissions, especially in a high wind speed regime (>10 m s$^{-1}$), which we discussed in lines 420-429. To clarify this point in the conclusion we changed the sentence to: "Furthermore, the most appropriate wind speed based k parameterization to compute air sea gas exchange, the main loss process for isoprene in the ocean, must be used in future studies."**

*Line 502: Could "The evaluation [...] should be examined" be worded differently?*

- **We changed the sentence starting at line 502 to: "Isoprene loss processes, in conjunction with the complexity of isoprene production, should be further examined in order to predict marine isoprene concentrations and evaluate the impact of isoprene on SOA formation over the remote open ocean."**

*Line 694 (Table 1): bold/italic is defined, but what are the R2 values that are neither?*

- **The authors do not state in their publications if these correlations are significant or not. We added this additional information to the table caption.**

*Fig 1: Why are not all station numbers shown? Where they are shown, it is often difficult to assign them to a particular dot. There also seem to be stations omitted or not visible? If they cannot be shown (same location as another one) or were not sampled (as suggested by Fig 3), please add this information to the caption. It may also be useful to add station numbers to Fig 3 to connect the two pieces of information.*

- **For a better readability we added not all but almost all station numbers to Figure 1 and added the following sentence to the figure caption: "Numbers indicate stations where a CTD depth profile was performed. Stations 6 & 8 (SPACES) as well as stations 4 & 6 and 13 & 14 (OASIS) have almost the same geographical coordinates. If a station number is omitted (SPACES: stations 5 & 7; OASIS: station 3, 5 & 12; ASTRA-OMZ: stations 4 & 9) no CTD cast was performed."**
**Station numbers are added to Figure 3.**

*Fig 5: Can you please show n in this figure for each set of data and add some details to the caption about the left vs. right part of the graph or refer to the main text (especially 5b) to clarify? Also, why are most of the whiskers for SPACES and OASIS in 5a different once the outliers have been excluded (other values should not be affected if one point is removed)? (For 5b, the new calculations can explain the changed whiskers, but are only mentioned in the main text.)*

- **We updated Figure 5 by showing the number of stations that were included for each set of data in the boxplot and provided some information in the figure caption: "Percent differences […] for the different cruises / cruise regions. Left of the vertical black line data is divided into the three different cruises, right of the vertical black line data is shown for the three cruises where outliers from left part are excluded. Additionally, ASTRA-OMZ was split into three regions (equator, coast, open ocean). Number of stations (n) used for each set of data is shown in italics. The red line represents the median, the boxes show the first to third quartile and the whiskers illustrate the highest and lowest values that are not outliers. The red plus signs represent outliers. The number indicated after \ denotes a station that has been excluded from the analysis."**
  **The referee is absolutely right, the whiskers should not be affected for SPACES and OASIS in Figure 5a when excluding the outliers. Accidently, the data for SPACES\1 and OASIS\10 in Figure 5a were interchanged. We have now fixed the figure.**

*Fig 6, 7, 8, 10: What do the error bars show? Error on measurement or standard deviation of the average? Please add this information to the caption.*

- **Error bars show the standard deviation of the average. This information was added to the figure captions.**

*Fig S2: Why was EdPAR(0+) calculated if there were also measurements available (binned data implies measured)?*

- **Measurements were not available for all stations, therefore EdPAR(0+) was calculated and verified with stations where measurements were available.**

*Fig S3: Why are chlorophytes and cyanobacteria functions not shown (EFs are listed in Table 2)? Please add to plot or add reason to caption.*

- **We added chlorophytes and cyanobacteria to figure S3.**

*Technical comments*

*Line 49: Change to "the concentrations generally range", as the following sentence presents different concentrations.*

- **Done.**

*Lines 76 and 454: reference should be Acuña Alvarez*

- **Done.**

*Line 131: Use "Phytoplankton functional types..." as heading for consistency*

- **Done.**

*Lines 133, 146 and 150: Change to "same stations as isoprene was sampled"; "subsurface irradiation", to define EdPAR(0-); and to "...the total chl-a concentration integrated..."*

- **Done.**

*Line 139/140: Replace "By that" with something like "This was used to derive..." or "The chl-a concs... were derived that way"*

- **Done.**

*Line 143 etc: Can PAR stand for both photosynthetically active radiation and photosynthetic available radiation? The latter does not seem commonly used.*

- **Yes, it can. In our manuscript we use "photosynthetic available radiation" consistently.**

*Line 163: EdPAR(0+) should have superscript and be in italics? (also in Fig S2?)*

- **Done.**

*Line 167: Suggest changing to "...due to aˇn shallow mixed layer depth (MLD) resulting in only one..."*

- **Done.**

*Line 254-256: Either the numbers or the description appears to be the wrong way round; dividing the mean by the concentration at a certain depth would give >1 for a smaller specific concentration.*

- **Fixed the description to "…we normalized the measured values by dividing the concentration of each depth of each station by the mean concentration in the mixed layer from the same station profile."**

*Lines 300, 318, 453: punctuation before "2)" is almost invisible; remove comma after "which"; add comma after halocarbons*

- **Done.**

*Line 308/318: Is there a difference between >80% of "total PFTs" and "total phytoplankton chl-a"? If not, this statement is only needed once.*

- **There is no difference and the second statement (line 318) was deleted.**

*Line 334, 357, 487: change "than" to "from"; "stations"; "in-field production rates"*

- **Done.**

*Line 388: "more saline" or "higher salinity"*

- **Done.**

*Line 441: Add "Here, [the loss rate constant...]" to start of the sentence to clarify.*

- **Done.**

*Line 499: must be further assessed? Furthermore, air-sea [...] has to be assessed?*

- **Done.**

*Line 504: evaluate "their" impact (of the isoprene concentrations - if this refers in fact to the evaluation of the processes, the sentence is not very clear and should be reworded)*

- **We changed the sentence to: "Isoprene loss processes, in conjunction with the complexity of isoprene production, should be further examined in order to predict marine isoprene concentrations and evaluate the impact of isoprene on SOA formation over the remote open ocean."**

*Line 507: A link to the database would be useful.*

- **As there is no data uploaded yet, we cannot provide a link, unfortunately. We will update as soon as possible.**

*Lines 704 and 738: (Table 3 and Fig 5 captions): remove the first "that"*

- **Done.**

*Fig 1: x-axis values partially obscured for OASIS/SPACES*

- **Done.**

*Fig 4 and Line 252 / Fig 8 and Lines 417-434: A darker shade of green would be easier to see (Fig 4); dotted lines are quite faint and legend covers error bar (Fig 8). Legend and description duplicate the information needed, details are also not needed in main text. ASTRA-OMZ details are also already given above the plot; check (c/d/e) (Fig 4).*

- **Done.**

*Fig 6 caption: Pchloronew , not Pchloro , according to main text?*

- **Done.**

*Fig S1: y-axis is umol m-2 s-1, while caption refers to W m-2. If a conversion was made, please specify.*

- **Done.**

**References**

[revised manuscript text omitted]

---

## Referee Report (RR1)

I have the following observations about the revised submission of the manuscript by Booge et al, to BGD.

1. L558-560: Moreover, our data leads to the conclusion that isoprene production rates in the field, irrespective of phytoplankton communities and their abundance, are influenced by salinity and nutrient levels, which has never been shown before.

   **Not quite. I urge the authors to consult Rinnan et al. 2014 (PCE) and the references within it, where they review the general point on how isoprene and other volatile emission is influenced by salinity in terrestrial and marine environments. There are several other papers that show a clear link between salinity levels and the extent of volatile emission (halogenated HCs in particular). Isoprene is just one among those volatiles.**

2. L405 to 418 in the revised version

   **Throughout the manuscript, the authors have emphasized the importance of differences between species in their responses to changes in temperature, light and other factors. For instance, *Prochlorococcus* is undoubtedly important purely because of its abundance that too in tropical waters. This is not a point of contention. The only thing the authors need to worry about is whether these species that are showing differences in the lab (accumulated from various studies), which the authors used to calculate their EFs, whether they are relevant in the global scheme for marine isoprene. I wish to see a statement somewhere that acknowledges the limitation of pooling emission responses within a PFT, without adequate consideration to their global abundance and relevance. If the authors are keen on species specific differences, then they must have some idea of the species they encountered and not just the PFT during these specific cruises they undertook. Kindly see that any emphasis on the differences between species, cuts both ways. The authors missed the point I made on this, during the first round of review.**

3. L438

   **Something odd has crept in L438. There are several places where there are some odd words or missing punctuation that make comprehension difficult. Please sieve the manuscript to get rid of such errors.**

4. L519-521 "This is a high isoprene production rate and we could assume higher isoprene concentrations higher concentrations of haptophytes. This relationship, however, is not evident (data not shown), which may be attributable to other processes masking this relationship"

   **These sentences do not read right. There is clearly something missing or wrong. Kindly revise.**

5. On bacterial degradation of isoprene

   **I am with the authors when they argue/speculate that isoprene degradation in the oceans can be due to bacterial consumption. It will just be a question of scale. Perhaps, haptophytes even use isoprene to attract bacteria to sustain their heterotrophic lifestyle. But, the evidence is not straight forward, at least not in this paper. Therefore, it is desirable that the phrase 'attributed to bacteria' in the abstract be replaced with "potentially due to degradation or consumption by bacteria". I believe some experiments are in their concluding phase in some labs, and there may soon be some evidence to directly say how bacteria can degrade/consume isoprene.**

---

## Author Response (AR2)

**We thank the referee for the comments which help to further improve the manuscript. We will address the comments in the following (bold). The lines refer to the revised uploaded manuscript.**

*L558-560: "Moreover, our data leads to the conclusion that isoprene production rates in the field, irrespective of phytoplankton communities and their abundance, are influenced by salinity and nutrient levels, which has never been shown before." Not quite. I urge the authors to consult Rinnan et al. 2014 (PCE) and the references within it, where they review the general point on how isoprene and other volatile emission is influenced by salinity in terrestrial and marine environments. There are several other papers that show a clear link between salinity levels and the extent of volatile emission (halogenated HCs in particular). Isoprene is just one among those volatiles.*

- **We thank the reviewer for this helpful information and changed the sentence starting in line 558 as follows:" Moreover, our data leads to the conclusion that isoprene production rates in the field, irrespective of phytoplankton communities and their abundance, are influenced by nutrient levels, which has never been shown before. Additionally, we show that isoprene production rates are influenced by salinity levels, which has also been observed in previous studies (Rinnan et al., 2014 and references therein).".**

*L405 to 418 in the revised version: Throughout the manuscript, the authors have emphasized the importance of differences between species in their responses to changes in temperature, light and other factors. For instance, Prochlorococcus is undoubtedly important purely because of its abundance that too in tropical waters. This is not a point of contention. The only thing the authors need to worry about is whether these species that are showing differences in the lab (accumulated from various studies), which the authors used to calculate their EFs, whether they are relevant in the global scheme for marine isoprene. I wish to see a statement somewhere that acknowledges the limitation of pooling emission responses within a PFT, without adequate consideration to their global abundance and relevance. If the authors are keen on species specific differences, then they must have some idea of the species they encountered and not just the PFT during these specific cruises they undertook. Kindly see that any emphasis on the differences between species, cuts both ways. The authors missed the point I made on this, during the first round of review.*

- **We now understand the point the reviewer wanted to make. Therefore, we added the following sentence in the method section 2.7.1, when introducing the emission factor (EF) of the different PFTs: "It should be noted that we are not sure what species were actually present during the cruises. We realize, therefore, that this method of calculating EFs is limited." Additionally, we changed the sentence starting in line 416 to: "Because we do not know which species were present, we hypothesize that the production rate is not dependent on one environmental parameter and varies from species to species within the group of cyanobacteria."**

*L438 : Something odd has crept in L438. There are several places where there are some odd words or missing punctuation that make comprehension difficult. Please sieve the manuscript to get rid of such errors.*

- **Thanks for pointing that out. We went through the manuscript and changed typos and punctuation errors.**

*L519-521: "This is a high isoprene production rate and we could assume higher isoprene concentrations higher concentrations of haptophytes. This relationship, however, is not evident (data not shown), which may be attributable to other processes masking this relationship". These sentences do not read right. There is clearly something missing or wrong. Kindly revise.*

- **We changed the sentence starting in line 519 to: "This is a high isoprene production rate and we could assume higher isoprene concentrations with higher concentrations of haptophytes. This relationship, however, is not evident (data not shown), which may indicate that other processes mask this relationship."**

*On bacterial degradation of isoprene: I am with the authors when they argue/speculate that isoprene degradation in the oceans can be due to bacterial consumption. It will just be a question of scale. Perhaps, haptophytes even use isoprene to attract bacteria to sustain their heterotrophic lifestyle. But, the evidence is not straight forward, at least not in this paper. Therefore, it is desirable that the phrase 'attributed to bacteria' in the abstract be replaced with "potentially due to degradation or consumption by bacteria". I believe some experiments are in their concluding phase in some labs, and there may soon be some evidence to directly say how bacteria can degrade/consume isoprene.*

[revised manuscript text omitted]